# Differentially Private Subspace Clustering

**Yining Wang, Yu-Xiang Wang and Aarti Singh**
Machine Learning Department, Carnegie Mellon Universty, Pittsburgh, USA
{yiningwa,yuxiangw,aarti}@cs.cmu.edu

## Abstract

Subspace clustering is an unsupervised learning problem that aims at grouping data points into multiple "clusters" so that data points in a single cluster lie approximately on a low-dimensional linear subspace. It is originally motivated by 3D motion segmentation in computer vision, but has recently been generically applied to a wide range of statistical machine learning problems, which often involves sensitive datasets about human subjects. This raises a dire concern for data privacy. In this work, we build on the framework of *differential privacy* and present two provably private subspace clustering algorithms. We demonstrate via both theory and experiments that one of the presented methods enjoys formal privacy and utility guarantees; the other one asymptotically preserves differential privacy while having good performance in practice. Along the course of the proof, we also obtain two new provable guarantees for the agnostic subspace clustering and the graph connectivity problem which might be of independent interests.

## 1 Introduction

Subspace clustering was originally proposed to solve very specific computer vision problems having a union-of-subspace structure in the data, e.g., motion segmentation under an affine camera model [11] or face clustering under Lambertian illumination models [15]. As it gains increasing attention in the statistics and machine learning community, people start to use it as an agnostic learning tool in social network [5], movie recommendation [33] and biological datasets [19]. The growing applicability of subspace clustering in these new domains inevitably raises the concern of *data privacy*, as many such applications involve dealing with sensitive information. For example, [19] applies subspace clustering to identify diseases from personalized medical data and [33] in fact uses subspace clustering as a effective tool to conduct linkage attacks on individuals in movie rating datasets. Nevertheless, privacy issues in subspace clustering have been less explored in the past literature, with the only exception of a brief analysis and discussion in [29]. However, the algorithms and analysis presented in [29] have several notable deficiencies. For example, data points are assumed to be incoherent and it only protects the differential privacy of any feature of a user rather than the entire user profile in the database. The latter means it is possible for an attacker to infer with high confidence whether a particular user is in the database, given sufficient side information.

It is perhaps reasonable why there is little work focusing on private subspace clustering, which is by all means a challenging task. For example, a negative result in [29] shows that if utility is measured in terms of exact clustering, then no private subspace clustering algorithm exists when neighboring databases are allowed to differ on an entire user profile. In addition, state-of-the-art subspace clustering methods like Sparse Subspace Clustering (SSC, [11]) lack a complete analysis of its clustering output, thanks to the notorious "graph connectivity" problem [21]. Finally, clustering could have high global sensitivity even if only cluster centers are released, as depicted in Figure 1. As a result, general private data releasing schemes like output perturbation [7, 8, 2] do not apply.

In this work, we present a systematic and principled treatment of differentially private subspace clustering. To circumvent the negative result in [29], we use the perturbation of recovered low-

dimensional subspace from the ground truth as the utility measure. Our contributions are two-fold. First, we analyze two efficient algorithms based on the sample-aggregate framework [22] and established formal privacy and utility guarantees when data are generated from some stochastic model or satisfy certain deterministic separation conditions. New results on (non-private) subspace clustering are obtained along our analysis, including a *fully agnostic* subspace clustering on well-separated datasets using stability arguments and *exact clustering* guarantee for thresholding-based subspace clustering (TSC, [14]) in the noisy setting. In addition, we employ the exponential mechanism [18] and propose a novel Gibbs sampler for sampling from this distribution, which involves a novel tweak in sampling from a matrix Bingham distribution. The method works well in practice and we show it is closely related to the well-known mixtures of probabilistic PCA model [27].

**Related work**   Subspace clustering can be thought as a generalization of PCA and $k$-means clustering. The former aims at finding a *single* low-dimensional subspace and the latter uses zero-dimensional subspaces as cluster centers. There has been extensive research on private PCA [2, 4, 10] and $k$-means [2, 22, 26]. Perhaps the most similar work to ours is [22, 4]. [22] applies the sample-aggregate framework to $k$-means clustering and [4] employs the exponential mechanism to recover private principal vectors. In this paper we give non-trivial generalization of both work to the private subspace clustering setting.

## 2   Preliminaries

### 2.1   Notations

For a vector $\boldsymbol{x} \in \mathbb{R}^d$, its $p$-norm is defined as $\|\boldsymbol{x}\|_p = (\sum_i \boldsymbol{x}_i^p)^{1/p}$. If $p$ is not explicitly specified then the 2-norm is used. For a matrix $\mathbf{A} \in \mathbb{R}^{n \times m}$, we use $\sigma_1(\mathbf{A}) \geq \cdots \geq \sigma_n(\mathbf{A}) \geq 0$ to denote its singular values (assuming without loss of generality that $n \leq m$). We use $\|\cdot\|_\xi$ to denote matrix norms, with $\xi = 2$ the matrix spectral norm and $\xi = F$ the Frobenious norm. That is, $\|\mathbf{A}\|_2 = \sigma_1(\mathbf{A})$ and $\|\mathbf{A}\|_F = \sqrt{\sum_{i=1}^n \sigma_i(\mathbf{A})^2}$. For a $q$-dimensional subspace $\mathcal{S} \subseteq \mathbb{R}^d$, we associate with a basis $\mathbf{U} \in \mathbb{R}^{d \times q}$, where the $q$ columns in $\mathbf{U}$ are orthonormal and $\mathcal{S} = \text{range}(\mathbf{U})$. We use $\mathbb{S}_q^d$ to denote the set of all $q$-dimensional subspaces in $\mathbb{R}^d$.

Given $\boldsymbol{x} \in \mathbb{R}^d$ and $\mathcal{S} \subseteq \mathbb{R}^d$, the distance $d(\boldsymbol{x}, \mathcal{S})$ is defined as $d(\boldsymbol{x}, \mathcal{S}) = \inf_{\boldsymbol{y} \in \mathcal{S}} \|\boldsymbol{x} - \boldsymbol{y}\|_2$. If $\mathcal{S}$ is a subspace associated with a basis $\mathcal{U}$, then we have $d(\boldsymbol{x}, \mathcal{S}) = \|\boldsymbol{x} - \mathcal{P}_\mathcal{S}(\boldsymbol{x})\|_2 = \|\boldsymbol{x} - \mathbf{U}\mathbf{U}^\top \boldsymbol{x}\|_2$, where $\mathcal{P}_\mathcal{S}(\cdot)$ denotes the projection operator onto subspace $\mathcal{S}$. For two subspaces $\mathcal{S}, \mathcal{S}'$ of dimension $q$, the distance $d(\mathcal{S}, \mathcal{S}')$ is defined as the Frobenious norm of the sin matrix of principal angles; i.e.,

$$d(\mathcal{S}, \mathcal{S}') = \|\sin\boldsymbol{\Theta}(\mathcal{S}, \mathcal{S}')\|_F = \|\mathbf{U}\mathbf{U}^\top - \mathbf{U}'\mathbf{U}'^\top\|_F, \tag{1}$$

where $\mathbf{U}, \mathbf{U}'$ are orthonormal basis associated with $\mathcal{S}$ and $\mathcal{S}'$, respectively.

### 2.2   Subspace clustering

Given $n$ data points $\boldsymbol{x}_1, \cdots, \boldsymbol{x}_n \in \mathbb{R}^d$, the task of subspace clustering is to cluster the data points into $k$ clusters so that data points within a subspace lie approximately on a low-dimensional subspace. Without loss of generality, we assume $\|\boldsymbol{x}_i\|_2 \leq 1$ for all $i = 1, \cdots, n$. We also use $\mathcal{X} = \{\boldsymbol{x}_1, \cdots, \boldsymbol{x}_n\}$ to denote the dataset and $\mathbf{X} \in \mathbb{R}^{d \times n}$ to denote the data matrix by stacking all data points in columnwise order. Subspace clustering seeks to find $k$ $q$-dimensional subspaces $\hat{\mathcal{C}} = \{\hat{\mathcal{S}}_1, \cdots, \hat{\mathcal{S}}_k\}$ so as to minimize the Wasserstein's distance or distance squared defined as

$$d_W^2(\hat{\mathcal{C}}, \mathcal{C}^*) = \min_{\pi:[k]\to[k]} \sum_{i=1}^k d^2(\hat{\mathcal{S}}_i, \mathcal{S}_{\pi(i)}^*), \tag{2}$$

where $\pi$ are taken over all permutations on $[k]$ and $\mathcal{S}^*$ are the optimal/ground-truth subspaces. In a model based approach, $\mathcal{C}^*$ is fixed and data points $\{\boldsymbol{x}_i\}_{i=1}^n$ are generated either deterministically or stochastically from one of the ground-truth subspaces in $\mathcal{C}^*$ with noise corruption; for a completely agnostic setting, $\mathcal{C}^*$ is defined as the minimizer of the $k$-means subspace clustering objective:

$$\mathcal{C}^* := \text{argmin}_{\mathcal{C}=\{\mathcal{S}_1,\cdots,\mathcal{S}_k\}\subseteq\mathbb{S}_q^d}\text{cost}(\mathcal{C};\mathcal{X}) = \text{argmin}_{\mathcal{C}=\{\mathcal{S}_1,\cdots,\mathcal{S}_k\}\subseteq\mathbb{S}_q^d}\frac{1}{n}\sum_{i=1}^n\min_j d^2(\boldsymbol{x}_i, \mathcal{S}_j). \tag{3}$$

To simplify notations, we use $\Delta_k(\mathcal{X}) = \text{cost}(\mathcal{C}^*;\mathcal{X})$ to denote cost of the optimal solution.

**Algorithm 1** The sample-aggregate framework [22]

1: **Input**: $\mathcal{X} = \{\boldsymbol{x}_i\}_{i=1}^n \subseteq \mathbb{R}^d$, number of subsets $m$, privacy parameters $\varepsilon, \delta$; $f, d_{\mathcal{M}}$.
2: **Initialize**: $s = \sqrt{m}$, $\alpha = \varepsilon/(5\sqrt{2\ln(2/\delta)})$, $\beta = \varepsilon/(4(D + \ln(2/\delta)))$.
3: **Subsampling**: Select $m$ random subsets of size $n/m$ of $\mathcal{X}$ independently and uniformly at random without replacement. Repeat this step until no single data point appears in more than $\sqrt{m}$ of the sets. Mark the subsampled subsets $\mathcal{X}_{S_1}, \cdots, \mathcal{X}_{S_m}$.
4: **Separate queries**: Compute $\mathcal{B} = \{\boldsymbol{s}_i\}_{i=1}^m \subseteq \mathbb{R}^D$, where $\boldsymbol{s}_i = f(\mathcal{X}_{S_i})$.
5: **Aggregation**: Compute $g(\mathcal{B}) = \boldsymbol{s}_{i^*}$ where $i^* = \operatorname{argmin}_{i=1}^m r_i(t_0)$ with $t_0 = (\frac{m+s}{2} + 1)$. Here $r_i(t_0)$ denotes the distance $d_{\mathcal{M}}(\cdot, \cdot)$ between $\boldsymbol{s}_i$ and the $t_0$-th nearest neighbor to $\boldsymbol{s}_i$ in $\mathcal{B}$.
6: **Noise calibration**: Compute $S(\mathcal{B}) = 2\max_k(\rho(t_0 + (k+1)s) \cdot e^{-\beta k})$, where $\rho(t)$ is the mean of the top $\lfloor s/\beta \rfloor$ values in $\{r_1(t), \cdots, r_m(t)\}$.
7: **Output**: $\mathcal{A}(\mathcal{X}) = g(\mathcal{B}) + \frac{S(\mathcal{B})}{\alpha}\boldsymbol{u}$, where $\boldsymbol{u}$ is a standard Gaussian random vector.

## 2.3 Differential privacy

**Definition 2.1** (Differential privacy, [7, 8]). *A randomized algorithm $\mathcal{A}$ is $(\varepsilon, \delta)$-differentially private if for all $\mathcal{X}, \mathcal{Y}$ satisfying $d(\mathcal{X}, \mathcal{Y}) = 1$ and all sets $S$ of possible outputs the following holds:*

$$\Pr[\mathcal{A}(\mathcal{X}) \in S] \leq e^\varepsilon \Pr[\mathcal{A}(\mathcal{Y}) \in S] + \delta. \tag{4}$$

*In addition, if $\delta = 0$ then the algorithm $\mathcal{A}$ is $\varepsilon$-differentially private.*

In our setting, the distance $d(\cdot, \cdot)$ between two datasets $\mathcal{X}$ and $\mathcal{Y}$ is defined as the number of different columns in $\mathbf{X}$ and $\mathbf{Y}$. Differential privacy ensures the output distribution is obfuscated to the point that every user has a plausible deniability about being in the dataset, and in addition any inferences about individual user will have nearly the same confidence before and after the private release.

# 3 Sample-aggregation based private subspace clustering

In this section we first summarize the sample-aggregate framework introduced in [22] and argue why it should be preferred to conventional output perturbation mechanisms [7, 8] for subspace clustering. We then analyze two efficient algorithms based on the sample-aggregate framework and prove formal privacy and utility guarantees. We also prove new results in our analysis regarding the stability of $k$-means subspace clustering (Lem. 3.3) and graph connectivity (i.e., consistency) of noisy threshold-based subspace clustering (TSC, [14]) under a stochastic model (Lem. 3.5).

## 3.1 Smooth local sensitivity and the sample-aggregate framework

Most existing privacy frameworks [7, 8] are based on the idea of *global sensitivity*, which is defined as the maximum output perturbation $\|f(\mathcal{X}_1) - f(\mathcal{X}_2)\|_\xi$, where maximum is over all neighboring databases $\mathcal{X}_1, \mathcal{X}_2$ and $\xi = 1$ or 2. Unfortunately, global sensitivity of clustering problems is usually high even if only cluster centers are released. For example, Figure 1 shows that the global sensitivity of $k$-means subspace clustering could be as high as $O(1)$, which ruins the algorithm utility.

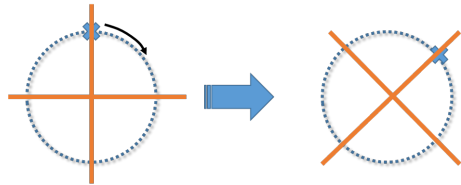

Figure 1: Illustration of instability of $k$-means subspace clustering solutions ($d = 2, k = 2, q = 1$). Blue dots represent evenly spaced data points on the unit circle; blue crosses indicate an additional data point. Red lines are optimal solutions.

To circumvent the above-mentioned challenges, Nissim et al. [22] introduces the sample-aggregate framework based on the concept of a smooth version of *local sensitivity*. Unlike global sensitivity, local sensitivity measures the maximum perturbation $\|f(\mathcal{X}) - f(\mathcal{X}')\|_\xi$ over all databases $\mathcal{X}'$ neighboring to the *input* database $\mathcal{X}$. The proposed sample-aggregate framework (pseudocode in Alg. 1) enjoys local sensitivity and comes with the following guarantee:

**Theorem 3.1** ([22], Theorem 4.2). *Let $f : \mathbb{D} \to \mathbb{R}^D$ be an efficiently computable function where $\mathbb{D}$ is the collection of all databases and $D$ is the output dimension. Let $d_{\mathcal{M}}(\cdot, \cdot)$ be a semimetric on*

the outer space of $f$. [1] *Set $\varepsilon > 2D/\sqrt{m}$ and $m = \omega(\log^2 n)$. The sample-aggregate algorithm $\mathcal{A}$ in Algorithm 1 is an efficient $(\varepsilon, \delta)$-differentially private algorithm. Furthermore, if $f$ and $m$ are chosen such that the $\ell_1$ norm of the output of $f$ is bounded by $\Lambda$ and*

$$\Pr_{\mathcal{X}_S \subseteq \mathcal{X}} [d_{\mathcal{M}}(f(\mathcal{X}_S), \boldsymbol{c}) \leq r] \geq \frac{3}{4} \tag{5}$$

*for some $\boldsymbol{c} \in \mathbb{R}^D$ and $r > 0$, then the standard deviation of Gaussian noise added is upper bounded by $O(r/\varepsilon) + \frac{\Lambda}{\varepsilon} e^{-\Omega(\frac{\varepsilon\sqrt{m}}{D})}$. In addition, when $m$ satisfies $m = \omega(D^2 \log^2(r/\Lambda)/\varepsilon^2)$, with high probability each coordinate of $\mathcal{A}(\mathcal{X}) - \bar{\boldsymbol{c}}$ is upper bounded by $O(r/\varepsilon)$, where $\bar{\boldsymbol{c}}$ depending on $\mathcal{A}(\mathcal{X})$ satisfies $d_{\mathcal{M}}(\boldsymbol{c}, \bar{\boldsymbol{c}}) = O(r)$.*

Let $f$ be any subspace clustering solver that outputs $k$ estimated low-dimensional subspaces and $d_{\mathcal{M}}$ be the Wasserstein's distance as defined in Eq. (2). Theorem 3.1 provides privacy guarantee for an efficient meta-algorithm with any $f$. In addition, utility guarantee holds with some more assumptions on input dataset $\mathcal{X}$. In following sections we establish utility guarantees. The main idea is to prove stability results as outlined in Eq. (5) for particular subspace clustering solvers and then apply Theorem 3.1.

### 3.2 The agnostic setting

We first consider the setting when data points $\{\boldsymbol{x}_i\}_{i=1}^n$ are arbitrarily placed. Under such agnostic setting the optimal solution $\mathcal{C}^*$ is defined as the one that minimizes the $k$-means cost as in Eq. (3). The solver $f$ is taken to be any $(1+\epsilon)$-approximation[2] of optimal $k$-means subspace clustering; that is, $f$ always outputs subspaces $\hat{\mathcal{C}}$ satisfying $\text{cost}(\hat{\mathcal{C}}; \mathcal{X}) \leq (1+\epsilon)\text{cost}(\mathcal{C}^*; \mathcal{X})$. Efficient core-set based approximation algorithms exist, for example, in [12]. The key task of this section it to identify assumptions under which the stability condition in Eq. (5) holds with respect to an approximate solver $f$. The example given in Figure 1 also suggests that identifiability issue arises when the input data $\mathcal{X}$ itself cannot be well clustered. For example, no two straight lines could well approximate data uniformly distributed on a circle. To circumvent the above-mentioned difficulty, we impose the following well-separation condition on the input data $\mathcal{X}$:

**Definition 3.2** (Well-separation condition for $k$-means subspace clustering). *A dataset $\mathcal{X}$ is $(\phi, \eta, \psi)$-well separated if there exist constants $\phi, \eta$ and $\psi$, all between 0 and 1, such that*

$$\Delta_k^2(\mathcal{X}) \leq \min\left\{\phi^2 \Delta_{k-1}^2(\mathcal{X}), \Delta_{k,-}^2(\mathcal{X}) - \psi, \Delta_{k,+}^2(\mathcal{X}) + \eta\right\}, \tag{6}$$

*where $\Delta_{k-1}, \Delta_{k,-}$ and $\Delta_{k,+}$ are defined as $\Delta_{k-1}^2(\mathcal{X}) = \min_{\mathcal{S}_{1:k-1}\in\mathbb{S}_q^d} \text{cost}(\{\mathcal{S}_i\}; \mathcal{X}); \Delta_{k,-}^2(\mathcal{X}) = \min_{\mathcal{S}_1\in\mathbb{S}_{q-1}^d, \mathcal{S}_{2:k}\in\mathbb{S}_q^d} \text{cost}(\{\mathcal{S}_i\}; \mathcal{X});$ and $\Delta_{k,+}^2(\mathcal{X}) = \min_{\mathcal{S}_1\in\mathbb{S}_{q+1}^d, \mathcal{S}_{2:k}\in\mathbb{S}_q^d} \text{cost}(\{\mathcal{S}_i\}; \mathcal{X}).$*

The first condition in Eq. (6), $\Delta_k^2(\mathcal{X}) \leq \phi^2 \Delta_{k-1}^2(\mathcal{X})$, constrains that the input dataset $\mathcal{X}$ cannot be well clustered using $k-1$ instead of $k$ clusters. It was introduced in [23] to analyze stability of $k$-means solutions. For subspace clustering, we need another two conditions regarding the intrinsic dimension of each subspace. The $\Delta_k^2(\mathcal{X}) \leq \Delta_{k,-}^2(\mathcal{X}) - \psi$ asserts that replacing a $q$-dimensional subspace with a $(q-1)$-dimensional one is not sufficient, while $\Delta_k^2(\mathcal{X}) \leq \Delta_{k,+}^2(\mathcal{X}) + \eta$ means an additional subspace dimension does not help much with clustering $\mathcal{X}$.

The following lemma is our main stability result for subspace clustering on well-separated datasets. It states that when a candidate clustering $\hat{\mathcal{C}}$ is close to the optimal clustering $\mathcal{C}^*$ in terms of clustering cost, they are also close in terms of the Wasserstein distance defined in Eq. (2).

**Lemma 3.3** (Stability of agnostic $k$-means subspace clustering). *Assume $\mathcal{X}$ is $(\phi, \eta, \psi)$-well separated with $\phi^2 < 1/1602$, $\psi > \eta$. Suppose a candidate clustering $\hat{\mathcal{C}} = \{\hat{\mathcal{S}}_1, \cdots, \hat{\mathcal{S}}_k\} \subseteq \mathbb{S}_q^d$ satisfies $\text{cost}(\hat{\mathcal{C}}; \mathcal{X}) \leq a \cdot \text{cost}(\mathcal{C}^*; \mathcal{X})$ for some $a < \frac{1-802\phi^2}{800\phi^2}$. Then the following holds:*

$$d_W(\hat{\mathcal{C}}, \mathcal{C}^*) \leq \frac{600\sqrt{2}\phi^2\sqrt{k}}{(1-150\phi^2)(\psi-\eta)}. \tag{7}$$

The following theorem is then a simple corollary, with a complete proof in Appendix B.

**Algorithm 2** Threshold-based subspace clustering (TSC), a simplified version

---
1: **Input**: $\mathcal{X} = \{\boldsymbol{x}_i\}_{i=1}^n \subseteq \mathbb{R}^d$, number of clusters $k$ and number of neighbors $s$.
2: **Thresholding**: construct $G \in \{0,1\}^{n \times n}$ by connecting $\boldsymbol{x}_i$ to the other $s$ data points in $\mathcal{X}$ with the largest absolute inner products $|\langle \boldsymbol{x}_i, \boldsymbol{x}' \rangle|$. Complete $G$ so that it is undirected.
3: **Clustering**: Let $\mathcal{X}^{(1)}, \cdots, \mathcal{X}^{(\ell)}$ be the connected components in $G$. Construct $\bar{\mathcal{X}}^{(\ell)}$ by sampling $q$ points from $\mathcal{X}^{(\ell)}$ uniformly at random without replacement.
4: **Output**: subspaces $\hat{\mathcal{C}} = \{\hat{\mathcal{S}}_{(\ell)}\}_{\ell=1}^k$; $\hat{\mathcal{S}}_{(\ell)}$ is the subspace spanned by $q$ arbitrary points in $\bar{\mathcal{X}}^{(\ell)}$.

---

**Theorem 3.4.** *Fix a $(\phi, \eta, \psi)$-well separated dataset $\mathcal{X}$ with $n$ data points and $\phi^2 < 1/1602$, $\psi > \eta$. Suppose $\mathcal{X}_S \subseteq \mathcal{X}$ is a subset of $\mathcal{X}$ with size $m$, sampled uniformly at random without replacement. Let $\hat{\mathcal{C}} = \{\hat{\mathcal{S}}_1, \cdots, \hat{\mathcal{S}}_2\}$ be an $(1+\epsilon)$-approximation of optimal $k$-means subspace clustering computed on $\mathcal{X}_S$. If $m = \Omega(\frac{kqd \log(qd/\gamma' \Delta_k^2(\mathcal{X}))}{\gamma'^2 \Delta_k^4(\mathcal{X})})$ with $\gamma' < \frac{1-802\phi^2}{800\phi^2} - 2(1+\epsilon)$, then we have:*

$$\Pr_{\mathcal{X}_S}\left[d_W(\hat{\mathcal{C}}, \mathcal{C}^*) \leq \frac{600\sqrt{2}\phi^2\sqrt{k}}{(1-150\phi^2)(\psi-\eta)}\right] \geq \frac{3}{4}, \tag{8}$$

*where $\mathcal{C}^* = \{\mathcal{S}_1^*, \cdots, \mathcal{S}_k^*\}$ is the optimal clustering on $\mathcal{X}$; that is, $\mathrm{cost}(\mathcal{C}^*; \mathcal{X}) = \Delta_k^2(\mathcal{X})$.*

Consequently, applying Theorem 3.4 together with the sample-aggregate framework we obtain a weak polynomial-time $\varepsilon$-differentially private algorithm for agnostic $k$-means subspace clustering, with additional amount of per-coordinate Gaussian noise upper bounded by $O(\frac{\phi^2\sqrt{k}}{\varepsilon(\psi-\eta)})$. Our bound is comparable to the one obtained in [22] for private $k$-means clustering, except for the $(\psi - \eta)$ term which characterizes the well-separatedness under the subspace clustering scenario.

### 3.3 The stochastic setting

We further consider the case when data points are stochastically generated from some underlying "true" subspace set $\mathcal{C}^* = \{\mathcal{S}_1^*, \cdots, \mathcal{S}_k^*\}$. Such settings were extensively investigated in previous development of subspace clustering algorithms [24, 25, 14]. Below we give precise definition of the considered stochastic subspace clustering model:

**The stochastic model**   For every cluster $\ell$ associated with subspace $\mathcal{S}_\ell^*$, a data point $\boldsymbol{x}_i^{(\ell)} \in \mathbb{R}^d$ belonging to cluster $\ell$ can be written as $\boldsymbol{x}_i^{(\ell)} = \boldsymbol{y}_i^{(\ell)} + \boldsymbol{\varepsilon}_i^{(\ell)}$, where $\boldsymbol{y}_i^{(\ell)}$ is sampled uniformly at random from $\{\boldsymbol{y} \in \mathcal{S}_\ell^* : \|\boldsymbol{y}\|_2 = 1\}$ and $\boldsymbol{\varepsilon}_i \sim \mathcal{N}(\boldsymbol{0}, \sigma^2/d \cdot \mathbf{I}_d)$ for some noise parameter $\sigma$.

Under the stochastic setting we consider the solver $f$ to be the Threshold-based Subspace Clustering (TSC, [14]) algorithm. A simplified version of TSC is presented in Alg. 2. An alternative idea is to apply results in the previous section since the stochastic model implies well-separated dataset when noise level $\sigma$ is small. However, the running time of TSC is $O(n^2 d)$, which is much more efficient than core-set based methods. TSC is provably correct in that the similarity graph $G$ has no false connections and is connected per cluster, as shown in the following lemma:

**Lemma 3.5** (Connectivity of TSC). *Fix $\gamma > 1$ and assume $\max 0.04 n_\ell \leq s \leq \min n_\ell/6$. If for every $\ell \in \{1, \cdots, k\}$, the number of data points $n_\ell$ and the noise level $\sigma$ satisfy*

$$\frac{n_\ell}{\log n_\ell} > \frac{\gamma \pi \sqrt{2q}(12\pi)^{q-1}}{0.01(q/2-1)(q-1)}; \quad \frac{\sigma(1+\sigma)}{\sqrt{\log n}}\frac{\sqrt{q}}{\sqrt{d}} \leq \frac{1}{15\log n} - \sqrt{1 - \min_{\ell \neq \ell'}\frac{d^2(\mathcal{S}_\ell^*, \mathcal{S}_{\ell'}^*)}{q}};$$

$$\bar{\sigma} < \sqrt{\frac{d}{24\log n}}\left[\cos\left(12\pi\left(\frac{\gamma\sqrt{2\pi q}\log n_\ell}{n_\ell}\right)^{\frac{1}{q-1}}\right) - \cos\left(\left(\frac{0.01(q/2-1)(q-1)}{\sqrt{\pi}}\right)^{\frac{1}{q-1}}\right)\right],$$

*where $\bar{\sigma} = 2\sqrt{5}\sigma + \sigma^2$. Then with probability at least $1 - n^2 e^{-\sqrt{d}} - n\sum_\ell e^{-n_\ell/400} - \sum_\ell n_\ell^{1-\gamma}/(\gamma \log n_\ell) - 12/n - \sum_\ell n_\ell e^{-c(n_\ell-1)}$, the connected components in $G$ correspond exactly to the $k$ subspaces.*

Conditions in Lemma 3.5 characterize the interaction between sample complexity $n_\ell$, noise level $\sigma$ and "signal" level $\min_{\ell \neq \ell'} d(\mathcal{S}_\ell^*, \mathcal{S}_{\ell'}^*)$. Theorem 3.6 is then a simple corollary of Lemma 3.5. Complete proofs are deferred to Appendix C.

**Theorem 3.6** (Stability of TSC on stochastic data). *Assume conditions in Lemma 3.5 hold with respect to $n' = n/m$ for $\omega(\log^2 n) \le m \le o(n)$. Assume in addition that $\lim_{n \to \infty} n_\ell = \infty$ for all $\ell = 1, \cdots, L$ and the failure probability does not exceed $1/8$. Then for every $\epsilon > 0$ we have*

$$\lim_{n \to \infty} \Pr_{\mathcal{X}_S} \left[ d_W(\hat{\mathcal{C}}, \mathcal{C}^*) > \epsilon \right] = 0. \tag{9}$$

Compared to Theorem 3.4 for the agnostic model, Theorem 3.6 shows that one can achieve *consistent* estimation of underlying subspaces under a stochastic model. It is an interesting question to derive finite sample bounds for the differentially private TSC algorithm.

### 3.4 Discussion

It is worth noting that the sample-aggregate framework is an $(\varepsilon, \delta)$-differentially private mechanism for any computational subroutine $f$. However, the utility claim (i.e., the $O(r/\varepsilon)$ bound on each coordinate of $\mathcal{A}(\mathcal{X}) - c$) requires the stability of the particular subroutine $f$, as outlined in Eq. (5). It is unfortunately hard to theoretically argue for stability of state-of-the-art subspace clustering methods such as sparse subspace cluster (SSC, [11]) due to the "graph connectivity" issue [21][3]. Nevertheless, we observe satisfactory performance of SSC based algorithms in simulations (see Sec. 5). It remains an open question to derive utility guarantee for (user) differentially private SSC.

## 4 Private subspace clustering via the exponential mechanism

In Section 3 we analyzed two algorithms with provable privacy and utility guarantees for subspace clustering based on the sample-aggregate framework. However, empirical evidence shows that sample-aggregate based private clustering suffers from poor utility in practice [26]. In this section, we propose a practical private subspace clustering algorithm based on the *exponential mechanism* [18]. In particular, given the dataset $\mathcal{X}$ with $n$ data points, we propose to samples parameters $\boldsymbol{\theta} = (\{\mathcal{S}_\ell\}_{\ell=1}^k, \{z_i\}_{i=1}^n)$ where $\mathcal{S}_\ell \in \mathbb{S}_d^q, z_j \in \{1, \cdots, k\}$ from the following distribution:

$$p(\boldsymbol{\theta}; \mathcal{X}) \propto \exp \left( -\frac{\varepsilon}{2} \cdot \sum_{i=1}^n d^2(\boldsymbol{x}_i, \mathcal{S}_{z_i}) \right), \tag{10}$$

where $\varepsilon > 0$ is the privacy parameter. The following proposition shows that exact sampling from the distribution in Eq. (10) results in a provable differentially private algorithm. Its proof is trivial and is deferred to Appendix D.1. Note that unlike sample-aggregate based methods, the exponential mechanism can privately release clustering assignment $z$. This does not violate the lower bound in [29] because the released clustering assignment $z$ is not guaranteed to be exactly correct.

**Proposition 4.1.** *The random algorithm $\mathcal{A} : \mathcal{X} \mapsto \boldsymbol{\theta}$ that outputs one sample from the distribution defined in Eq. (10) is $\varepsilon$-differential private.*

### 4.1 A Gibbs sampling implementation

It is hard in general to sample parameters from distributions as complicated as in Eq. (10). We present a Gibbs sampler that iteratively samples subspaces $\{\mathcal{S}_i\}$ and cluster assignments $\{z_j\}$ from their conditional distributions.

**Update of $z_i$:** When $\{\mathcal{S}_\ell\}$ and $z_{-i}$ are fixed, the conditional distribution of $z_i$ is

$$p(z_i | \{\mathcal{S}_\ell\}_{\ell=1}^k, z_{-i}; \mathcal{X}) \propto \exp(-\varepsilon/2 \cdot d^2(\boldsymbol{x}_i, \mathcal{S}_{z_i})). \tag{11}$$

Since $d(\boldsymbol{x}_i, \mathcal{S}_{z_i})$ can be efficiently computed (given an orthonormal basis of $\mathcal{S}_{z_i}$), update of $z_i$ can be easily done by sampling $z_j$ from a categorical distribution.

**Update of $\mathcal{S}_\ell$:** Let $\widetilde{\mathcal{X}}^{(\ell)} = \{\boldsymbol{x}_i \in \mathcal{X} : z_i = \ell\}$ denote data points that are assigned to cluster $\ell$ and $\tilde{n}_\ell = |\widetilde{\mathcal{X}}^{(\ell)}|$. Denote $\widetilde{\mathbf{X}}^{(\ell)} \in \mathbb{R}^{d \times \tilde{n}_\ell}$ as the matrix with columns corresponding to all data points in $\widetilde{\mathcal{X}}^{(\ell)}$. The distribution over $\mathcal{S}_\ell$ conditioned on $z$ can then be written as

$$p(\mathcal{S}_\ell = \text{range}(\mathbf{U}_\ell) | z; \mathcal{X}) \propto \exp(\varepsilon/2 \cdot \text{tr}(\mathbf{U}_\ell^\top \mathbf{A}_\ell \mathbf{U}_\ell)); \quad \mathbf{U}_\ell \in \mathbb{R}^{d \times q}, \mathbf{U}_\ell^\top \mathbf{U}_\ell = \mathbf{I}_{q \times q}, \tag{12}$$

where $\mathbf{A}_\ell = \widetilde{\mathbf{X}}^{(\ell)} \widetilde{\mathbf{X}}^{(\ell)^\top}$ is the unnormalized sample covariance matrix. Distribution of the form in Eq. (12) is a special case of the *matrix Bingham distribution*, which admits a Gibbs sampler [16]. We give implementation details in Appendix D.2 with modifications so that the resulting Gibbs sampler is empirically more efficient for a wide range of parameter settings.

## 4.2 Discussion

The proposed Gibbs sampler resembles the $k$-plane algorithm for subspace clustering [3]. It is in fact a "probabilistic" version of $k$-plane since sampling is performed at each iteration rather than deterministic updates. Furthermore, the proposed Gibbs sampler could be viewed as posterior sampling for the following generative model: first sample $\mathbf{U}_\ell$ uniformly at random from $\mathbb{S}_q^d$ for each subspace $\mathcal{S}_\ell$; afterwards, cluster assignments $\{z_i\}_{i=1}^n$ are sampled such that $\Pr[z_i = j] = 1/k$ and $\boldsymbol{x}_i$ is set as $\boldsymbol{x}_i = \mathbf{U}_\ell \boldsymbol{y}_i + \mathcal{P}_{\mathbf{U}_\ell^\perp} \boldsymbol{w}_i$, where $\boldsymbol{y}_i$ is sampled uniformly at random from the $q$-dimensional unit ball and $\boldsymbol{w}_i \sim \mathcal{N}(0, \mathbf{I}_d/\varepsilon)$. Connection between the above-mentioned generative model and Gibbs sampler is formally justified in Appendix D.3. The generative model is strikingly similar to the well-known mixtures of probabilistic PCA (MPPCA, [27]) model by setting variance parameters $\sigma_\ell$ in MPPCA to $\sqrt{1/\varepsilon}$. The only difference is that $\boldsymbol{y}_i$ are sampled uniformly at random from a unit ball [4] and noise $\boldsymbol{w}_i$ is constrained to $\mathbf{U}_\ell^\perp$, the complement space of $\mathbf{U}_\ell$. Note that this is closely related to earlier observation that "posterior sampling is private" [20, 6, 31], but different in that we constructed a model from a private procedure rather than the other way round.

As the privacy parameter $\varepsilon \to \infty$ (i.e., no privacy guarantee), we arrive immediately at the exact $k$-plane algorithm and the posterior distribution concentrates around the optimal $k$-means solution $(\mathcal{C}^*, \boldsymbol{z}^*)$. This behavior is similar to what a small-variance asymptotic analysis on MPPCA models reveals [30]. On the other hand, the proposed Gibbs sampler is significantly different from previous Bayesian probabilisitic PCA formulation [34, 30] in that the subspaces are sampled from a matrix Bingham distribution. Finally, we remark that the proposed Gibbs sampler is only asymptotically private because Proposition 4.1 requires exact (or nearly exact [31]) sampling from Eq. (10).

## 5 Numerical results

We provide numerical results of both the sample-aggregate and Gibbs sampling algorithms on synthetic and real-world datasets. We also compare with a baseline method implemented based on the $k$-plane algorithm [3] with perturbed sample covariance matrix via the SuLQ framework [2] (details presented in Appendix E). Three solvers are considered for the sample-aggregate framework: threshold-based subspace clustering (TSC, [14]), which has provable utility guarantee with sample-aggregation on stochastic models, along with sparse subspace clustering (SSC, [11]) and low-rank representation (LRR, [17]), the two state-of-the-art methods for subspace clustering. For Gibbs sampling, we use non-private SSC and LRR solutions as initialization for the Gibbs sampler. All methods are implemented using Matlab.

For synthetic datasets, we first generate $k$ random $q$-dimensional linear subspaces. Each subspace is generated by first sampling a $d \times q$ random Gaussian matrix and then recording its column space. $n$ data points are then assigned to one of the $k$ subspaces (clusters) uniformly at random. To generate a data point $\boldsymbol{x}_i$ assigned with subspace $\mathcal{S}_\ell$, we first sample $\boldsymbol{y}_i \in \mathbb{R}^q$ with $\|\boldsymbol{y}_i\|_2 = 1$ uniformly at random from the $q$-dimensional unit sphere. Afterwards, $\boldsymbol{x}_i$ is set as $\boldsymbol{x}_i = \mathbf{U}_\ell \boldsymbol{y}_i + \boldsymbol{w}_i$, where $\mathbf{U}_\ell \in \mathbb{R}^{d \times q}$ is an orthonormal basis associated with $\mathcal{S}_\ell$ and $\boldsymbol{w}_i \sim \mathcal{N}(\mathbf{0}, \sigma^2 \mathbf{I}_d)$ is a noise vector.

Figure 2 compares the utility (measured in terms of $k$-means objective $\mathrm{cost}(\hat{\mathcal{C}}; \mathcal{X})$ and the Wasserstein's distance $d_W(\hat{\mathcal{C}}, \mathcal{C}^*)$) of sample aggregation, Gibbs sampling and SuLQ subspace clustering. As shown in the plots, sample-aggregation algorithms have poor utility unless the privacy parameter $\varepsilon$ is truly large (which means very little privacy protection). On the other hand, both Gibbs sampling and SuLQ subspace clustering give reasonably good performance. Figure 2 also shows that SuLQ scales poorly with the ambient dimension $d$. This is because SuLQ subspace clustering requires calibrating noise to a $d \times d$ sample covariance matrix, which induces much error when $d$ is large. Gibbs sampling seems to be robust to various $d$ settings.

We also experiment on real-world datasets. The right two plots in Figure 2 report utility on a subset of the extended Yale Face Dataset B [13] for face clustering. 5 random individuals are picked, forming a subset of the original dataset with $n = 320$ data points (images). The dataset is preprocessed by projecting each individual onto a 9D affine subspace via PCA. Such preprocessing step was adopted in [32, 29] and was theoretically justified in [1]. Afterwards, ambient dimension of the entire dataset is reduced to $d = 50$ by random Gaussian projection. The plots show that Gibbs sampling significantly outperforms the other algorithms.

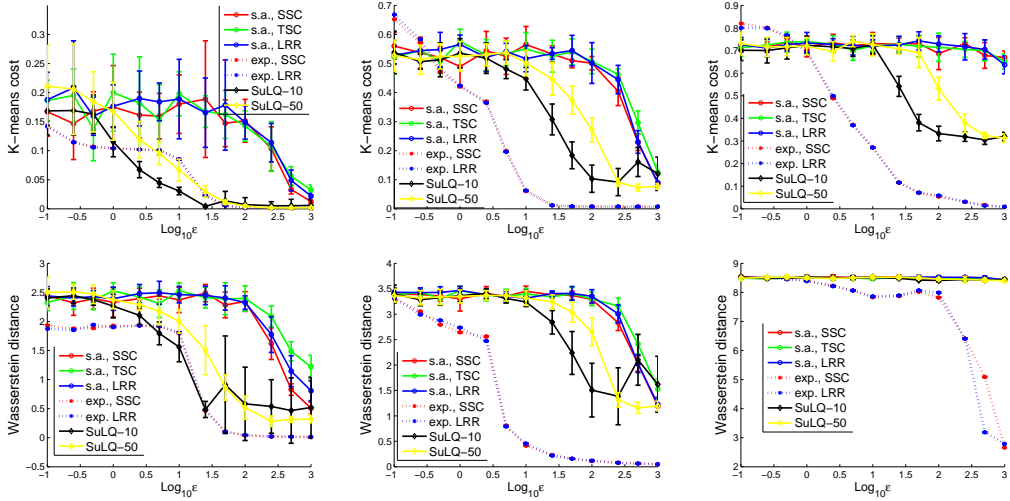

Figure 2: Utility under fixed privacy budget $\varepsilon$. Top row shows $k$-means cost and bottom row shows the Wasserstein's distance $d_W(\hat{\mathcal{C}}, \mathcal{C}^*)$. From left to right: synthetic dataset, $n = 5000, d = 5, k = 3, q = 3, \sigma = 0.01$; $n = 1000, d = 10, k = 3, q = 3, \sigma = 0.1$; extended Yale Face Dataset B (a subset). $n = 320, d = 50, k = 5, q = 9, \sigma = 0.01$. $\delta$ is set to $1/(n \ln n)$ for $(\varepsilon, \delta)$-privacy algorithms. "s.a." stands for smooth sensitivity and "exp." stands for exponential mechanism. "SuLQ-10" and "SuLQ-50" stand for the SuLQ framework performing 10 and 50 iterations. Gibbs sampling is run for 10000 iterations and the mean of the last 100 samples is reported.

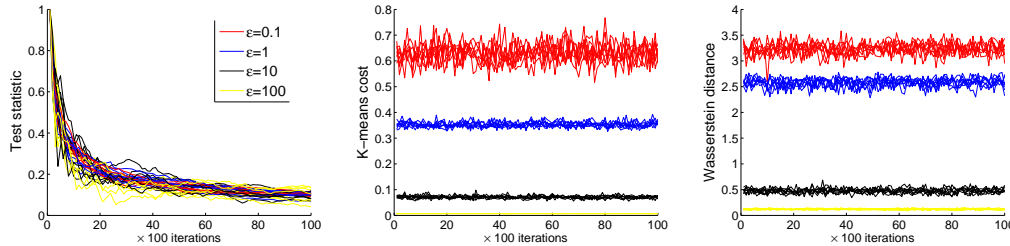

Figure 3: Test statistics, $k$-means cost and $d_W(\hat{\mathcal{C}}, \mathcal{C}^*)$ of 8 trials of the Gibbs sampler under different privacy settings. Synthetic dataset setting: $n = 1000, d = 10, k = 3, q = 3, \sigma = 0.1$.

In Figure 3 we investigate the mixing behavior of proposed Gibbs sampler. We plot for multiple trials of Gibbs sampling the $k$-means objective, Wasserstein's distance and a test statistic $1/\sqrt{kq} \cdot (\sum_{\ell=1}^{k} \|1/T \cdot \sum_{t=1}^{T} \mathbf{U}_{\ell}^{(t)}\|_F^2)^{1/2}$, where $\mathbf{U}_{\ell}^{(t)}$ is a basis sample of $\mathcal{S}_{\ell}$ at the $t$th iteration. The test statistic has mean zero under distribution in Eq. (10) and a similar statistic was used in [4] as a diagnostic of the mixing behavior of another Gibbs sampler. Figure 3 shows that under various privacy parameter settings, the proposed Gibbs sampler mixes quite well after 10000 iterations.

## 6   Conclusion

In this paper we consider subspace clustering subject to formal differential privacy constraints. We analyzed two sample-aggregate based algorithms with provable utility guarantees under agnostic and stochastic data models. We also propose a Gibbs sampling subspace clustering algorithm based on the exponential mechanism that works well in practice. Some interesting future directions include utility bounds for state-of-the-art subspace clustering algorithms like SSC or LRR.

**Acknowledgement**   This research is supported in part by grant NSF CAREER IIS-1252412, NSF Award BCS-0941518, and a grant by Singapore National Research Foundation under its International Research Centre @ Singapore Funding Initiative administered by the IDM Programme Office.

## Footnotes

[1] $d_{\mathcal{M}}(\cdot, \cdot)$ satisfies $d_{\mathcal{M}}(x, y) \geq 0$, $d_{\mathcal{M}}(x, x) = 0$ and $d_{\mathcal{M}}(x, y) \leq d_{\mathcal{M}}(x, z) + d_{\mathcal{M}}(y, z)$ for all $x, y, z$.

[2] Here $\epsilon$ is an approximation constant and is not related to the privacy parameter $\varepsilon$.

[3]Recently [28] established full clustering guarantee for SSC, however, under strong assumptions.

[4] In MPPCA latent variables $\boldsymbol{y}_i$ are sampled from a normal distribution $\mathcal{N}(\mathbf{0}, \rho^2 \mathbf{I}_q)$.

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
