[Supplementary Material]

# Appendix A    Some basic properties regarding the distances

**Proposition A.1.** *Let $\mathcal{S}_1, \mathcal{S}_2, \mathcal{S}_3 \in \mathbb{S}_q^d$ be three $q$-dimensional subspaces. Then $d(\mathcal{S}_1, \mathcal{S}_3) \leq d(\mathcal{S}_1, \mathcal{S}_2) + d(\mathcal{S}_2, \mathcal{S}_3)$ and $d^2(\mathcal{S}_1, \mathcal{S}_3) \leq 2(d^2(\mathcal{S}_1, \mathcal{S}_2) + d^2(\mathcal{S}_2, \mathcal{S}_3))$.*

*Proof.* Let $\mathbf{U}_1, \mathbf{U}_2, \mathbf{U}_3 \in \mathbb{R}^{d \times q}$ be orthonormal basis associated with $\mathcal{S}_1, \mathcal{S}_2$ and $\mathcal{S}_3$. We then have $d(\mathcal{S}_1, \mathcal{S}_3) = \|\mathbf{U}_1\mathbf{U}_1^\top - \mathbf{U}_3\mathbf{U}_3^\top\|_F \leq \|\mathbf{U}_1\mathbf{U}_1^\top - \mathbf{U}_2\mathbf{U}_2^\top\|_F + \|\mathbf{U}_2\mathbf{U}_2^\top - \mathbf{U}_3\mathbf{U}_3^\top\|_F = d(\mathcal{S}_1, \mathcal{S}_2) + d(\mathcal{S}_2, \mathcal{S}_3)$. The other inequality holds due to the fact that $\|\mathbf{U}_1\mathbf{U}_1^\top - \mathbf{U}_3\mathbf{U}_3^\top\|_F^2 \leq 2(\|\mathbf{U}_1\mathbf{U}_1^\top - \mathbf{U}_2\mathbf{U}_2^\top\|_F^2 + \|\mathbf{U}_2\mathbf{U}_2^\top - \mathbf{U}_3\mathbf{U}_3^\top\|_F^2)$. $\square$

**Proposition A.2.** *For any $\boldsymbol{x} \in \mathbb{R}^d$ and $\mathcal{S}, \mathcal{S}' \in \mathbb{S}_q^d$, we have $d(\boldsymbol{x}, \mathcal{S}') \leq d(\boldsymbol{x}, \mathcal{S}) + d(\mathcal{S}, \mathcal{S}')$ and $d^2(\boldsymbol{x}, \mathcal{S}') \leq 2(d^2(\boldsymbol{x}, \mathcal{S}) + d^2(\mathcal{S}, \mathcal{S}'))$.*

*Proof.* By definition, $d(\boldsymbol{x}, \mathcal{S}') = \|\boldsymbol{x} - \mathcal{P}_{\mathcal{S}'}(\boldsymbol{x})\|_2 \leq \|\boldsymbol{x} - \mathcal{P}_{\mathcal{S}'}(\mathcal{P}_{\mathcal{S}}(\boldsymbol{x}))\|_2 \leq \|\boldsymbol{x} - \mathcal{P}_{\mathcal{S}}(\boldsymbol{x})\|_2 + \|\mathcal{P}_{\mathcal{S}}(\boldsymbol{x}) - \mathcal{P}_{\mathcal{S}'}(\mathcal{P}_{\mathcal{S}}(\boldsymbol{x}))\|_2 \leq d(\boldsymbol{x}, \mathcal{S}) + \sup_{\boldsymbol{y} \in \mathcal{S}, \|\boldsymbol{y}\|_2 \leq 1} \|\boldsymbol{y} - \mathcal{P}_{\mathcal{S}'}(\boldsymbol{y})\|_2$. Note also that $\sup_{\boldsymbol{y} \in \mathcal{S}, \|\boldsymbol{y}\|_2 \leq 1} \|\boldsymbol{y} - \mathcal{P}_{\mathcal{S}'}(\boldsymbol{y})\|_2 \leq \sup_{\boldsymbol{y} \in \mathcal{S}, \|\boldsymbol{y}\|_2 \leq 1} \|\mathbf{U}\mathbf{U}^\top\boldsymbol{y} - \mathbf{U}'\mathbf{U}'^\top\boldsymbol{y}\|_2 \leq \|\mathbf{U}\mathbf{U}^\top - \mathbf{U}'\mathbf{U}'^\top\|_2 \leq d(\mathcal{S}, \mathcal{S}')$. Here $\mathcal{U}$ and $\mathcal{U}'$ are orthonormal basis associated with $\mathcal{S}$ and $\mathcal{S}'$. Therefore, $d(\boldsymbol{x}, \mathcal{S}') \leq d(\boldsymbol{x}, \mathcal{S}) + d(\mathcal{S}, \mathcal{S}')$. The other inequality follows by the same argument. $\square$

**Proposition A.3.** *Fix $\mathcal{S} \in \mathbb{S}_q^d$ and let $\mathbf{U} \in \mathbb{R}^{d \times q}$ be an orthonormal basis associated with $\mathcal{S}$. Suppose $\mathbf{U}' = \mathbf{U} + \mathbf{E}$ and $\mathcal{S}' = \text{range}(\mathbf{U}')$. Then $d(\mathcal{S}, \mathcal{S}') \leq \sqrt{2}\|\mathbf{E}\|_F$.*

*Proof.* Apply Wedin's Theorem (Theorem F.2 in Appendix F) and note that $\sigma_q(\mathbf{U}) = 1$. $\square$

# Appendix B    Proofs of sample-aggregate private subspace clustering: the agnostic case

The main objective of this section is to prove Theorem 3.4 for differentially private subspace clustering under the fully agnostic setting. The theorem is a simple consequence of Lemma 3.3 in the main text and the following lemma:

**Lemma B.1.** *Fix $\gamma > 0$. Suppose $\mathcal{X}_S$ contains $m = \Omega(\frac{kqd\log(qd/\gamma)}{\gamma^2})$ data points subsampled from $\mathcal{X}$ uniformly at random without replacement. Then with probability at least $3/4$ over random samples $U$, the following holds* uniformly *for all candidate subspace sets $\mathcal{C}$:*

$$\text{cost}(\mathcal{C}; \mathcal{X}_S) \leq 2\text{cost}(\mathcal{C}; \mathcal{X}) + \gamma. \tag{13}$$

## B.1    Proof of Lemma B.1

**Lemma B.2** ([39]). *Fix $\mathcal{X}$ and $f : \mathcal{X} \to [0, M]$ for some positive constant $M > 0$. Let $\mathcal{X}_S$ be a subset of $\mathcal{X}$ with $t$ elements, each drawn uniformly at random from $\mathcal{X}$ without replacement. Let $\epsilon, \delta > 0$. Then $\Pr[|\mathbb{E}_\mathcal{X}[f(x)] - \mathbb{E}_{\mathcal{X}_S}[f(x)]| \geq \epsilon] \leq \delta$ when $t \geq \frac{M^2 \ln(2/\delta)}{2\epsilon^2}$.*

**Corollary B.3.** *Fix $\mathcal{X}$ and a finite set of functions $\mathcal{F}$, where $0 \leq f(x) \leq M$ for every $x \in \mathcal{X}$ and $f \in \mathcal{F}$. Let $\mathcal{X}_S$ be a subset of $\mathcal{X}$ with $m$ elements, each drawn uniformly at random from $\mathcal{X}$ without replacement. Let $\epsilon, \delta > 0$. Then $\Pr[\exists f \in \mathcal{F}, |\mathbb{E}_\mathcal{X}[f(x)] - \mathbb{E}_{\mathcal{X}_S}[f(x)]| \geq \epsilon] \leq \delta$ when $m \geq \frac{M^2 \ln(2|\mathcal{F}|/\delta)}{2\epsilon^2}$.*

*Proof.* Apply Lemma B.2 and use union bound over all $f \in \mathcal{F}$. $\square$

**Lemma B.4.** *Fix $\epsilon > 0$. There exists $\mathbb{S} \subseteq \mathbb{S}_q^d$ with $|\mathbb{S}| = O((qd)^{qd/2}/\epsilon^{qd})$ such that for any $\mathcal{S} \in \mathbb{S}_q^d$, $\min_{\mathcal{S}' \in \mathbb{S}} d(\mathcal{S}, \mathcal{S}') \leq \epsilon$.*

*Proof.* By a standard convering number argument, there exists $\mathbb{L} \subseteq \mathbb{R}^d$ with $|\mathbb{L}| = O((\sqrt{d}/\epsilon)^d)$ such that for any $\boldsymbol{x} \in \mathbb{R}^d$, $\|\boldsymbol{x}\|_2 \leq 1$, we have $\min_{\boldsymbol{x}' \in \mathbb{L}} \|\boldsymbol{x} - \boldsymbol{x}'\|_2 \leq \epsilon$. Consequently, there exists $\mathbb{L}_q \subseteq \mathbb{R}^{d \times q}$ with $|\mathbb{L}_q| = O((qd)^{qd/2}/\epsilon^{qd})$ such that for any $\mathbf{U} \in \mathbb{R}^{d \times q}$ with unit column norms, $\min_{\mathbf{U}' \in \mathbb{L}_q} \|\mathbf{U} - \mathbf{U}'\|_F \leq \epsilon$. Proposition A.3 then yields the lemma. $\square$

We are now ready to prove Lemma B.1.

*Proof of Lemma B.1.* Suppose $\mathbb{S}$ is a finite subset of $\mathbb{S}_q^d$ such that for every $\mathcal{S} \in \mathbb{S}_q^d$, $\min_{\mathcal{S}' \in \mathbb{S}} d^2(\mathcal{S}, \mathcal{S}') \leq \gamma/4$. By Lemma B.4, there exists such $\mathbb{S}$ with $|\mathbb{S}| = O((qd/\gamma)^{qd/2})$. Let $\mathcal{X}$ be the set of data points and $\mathcal{F} = \{f(\cdot; \mathcal{C}) | \mathcal{C} = \{\mathcal{S}_1, \cdots, \mathcal{S}_k\} \subseteq \mathbb{S}\}$, where $f(\boldsymbol{x}; \mathcal{C}) = \min_{j=1}^k d^2(\boldsymbol{x}, \mathcal{S}_j)$. By definition, $\mathbb{E}_{\mathcal{X}}[f(\boldsymbol{x}; \mathcal{C})] = \mathrm{cost}(\mathcal{C}; \mathcal{X})$ and $|\mathcal{F}| = O((qd/\gamma)^{kqd/2})$. Subsequently, applying Corollary B.3 we obtain

$$\Pr_{\mathcal{X}_S}\left[\forall \mathcal{C} \subseteq \mathbb{S}, |\mathrm{cost}(\mathcal{C}; \mathcal{X}_S) - \mathrm{cost}(\mathcal{C}; \mathcal{X})| \leq \frac{\gamma}{2}\right] \geq \frac{3}{4}$$

whenever $|\mathcal{X}_S| = \Omega(\frac{kqd\log(qd/\gamma)}{\gamma^2})$. Consequently, applying Proposition A.2 we have

$$\Pr_{\mathcal{X}_S}\left[\forall \mathcal{C} \subseteq \mathbb{S}_q^d, \mathrm{cost}(\mathcal{C}; \mathcal{X}_S) \leq 2\mathrm{cost}(\mathcal{C}; \mathcal{X}) + \gamma\right] \geq \frac{3}{4}.$$

$\square$

## B.2 Proof of Lemma 3.3

We first define some notations that will be used in the proof. Throughout the section we assume the dataset $\mathcal{X}$ is $(\phi, \eta, \psi)$-well separated. Let $\mathcal{X}_i = \{\boldsymbol{x} \in \mathcal{X} : d(\boldsymbol{x}, \mathcal{S}_i^*) \leq d(\boldsymbol{x}, \mathcal{S}_j^*), \forall j\}$ denote the collection of all data points in $\mathcal{X}$ that are clustered to the cluster corresponding to $\mathcal{S}_i^*$. Define $n_i = |\mathcal{X}_i|$. By definition, $\sum_{i=1}^k n_i = n$. Define $r_i^2 = \Delta_1^2(\mathcal{X}_i)$, $D_i = \min_{j \neq i} d(\mathcal{S}_i^*, \mathcal{S}_j^*)$ and $d_i^2 = \phi^2 n \Delta_{k-1}^2(\mathcal{X})/n_i$. Let $\mathcal{X}_i^{\mathrm{cor}} = \{\boldsymbol{x} \in \mathcal{X}_i : d(\boldsymbol{x}, \mathcal{S}_i^*)^2 \leq \frac{r_i^2}{\rho}\}$ for some parameter $\rho \in (0, 1)$.

**Proposition B.5.** $r_i^2 \leq d_i^2 \leq \frac{2\phi^2}{1-2\phi^2} D_i^2$.

*Proof.* Since $\mathcal{X}$ is well-separated we have $d_i^2 = \phi^2 \Delta_{k-1}^2(\mathcal{X}) \cdot n/n_i \geq \Delta_k^2(\mathcal{X}) \cdot n/n_i \geq \frac{1}{n_i} \sum_{\boldsymbol{x} \in \mathcal{X}_i} d(\boldsymbol{x}, \mathcal{S}_i^*)^2 \geq \Delta_1^2(\mathcal{X}_i) = r_i^2$. Hence the first inequality.

For the second inequality, we only need to prove that $(1 - 2\phi^2)n\Delta_{k-1}^2(\mathcal{X}) \leq 2n_i D_i^2$. By well-separatedness $(1 - 2\phi^2)n\Delta_{k-1}^2(\mathcal{X}) = n(\Delta_{k-1}^2(\mathcal{X}) - 2\Delta_k^2(\mathcal{X}))$. On the other hand, by diverting all points in $\mathcal{X}_i^*$ into the cluster associated with $\mathcal{S}_j^*$ with $d(\mathcal{S}_i^*, \mathcal{S}_j^*) = D_i$, we have

$$
\begin{aligned}
n\Delta_{k-1}^2(\mathcal{X}) &\leq \sum_{\ell \neq i} \sum_{\boldsymbol{x} \in \mathcal{X}_\ell} d^2(\boldsymbol{x}, \mathcal{S}_\ell^*) + \sum_{\boldsymbol{x} \in \mathcal{X}_i} d^2(\boldsymbol{x}, \mathcal{S}_j^*) \\
&\leq \sum_{\ell \neq i} \sum_{\boldsymbol{x} \in \mathcal{X}_\ell} d^2(\boldsymbol{x}, \mathcal{S}_\ell^*) + 2 \sum_{\boldsymbol{x} \in \mathcal{X}_i} d^2(\boldsymbol{x}, \mathcal{S}_i^*) + 2n_i d^2(\mathcal{S}_i^*, \mathcal{S}_j^*) \\
&\leq 2n\Delta_k^2(\mathcal{X}) + 2n_i D_i^2.
\end{aligned}
$$

Rearranging the terms we get $n(\Delta_{k-1}^2(\mathcal{X}) - 2\Delta_k^2(\mathcal{X})) \leq 2n_i D_i^2$. $\square$

**Proposition B.6.** *For any $\rho \in (0, 1)$, $|\mathcal{X}_i^{\mathrm{cor}}| \geq (1 - \rho)|\mathcal{X}_i| = (1 - \rho)n_i$.*

*Proof.* By definition $r_i^2 = \frac{1}{n_i} \sum_{\boldsymbol{x} \in \mathcal{X}_i} d^2(\boldsymbol{x}, \mathcal{S}_i^*) = \mathbb{E}[T]$, where $T$ is the random variable of $d^2(\boldsymbol{x}, \mathcal{S}_i^*)$ for a vector chosen from $\mathcal{X}_i$ uniformly at random. By Markov's inequality, $\Pr[T > \frac{r_i^2}{\rho}] \leq \rho$ and hence $|\mathcal{X}_i^{\mathrm{cor}}| = n_i \Pr[T \leq \frac{r_i^2}{\rho}] \geq (1 - \rho)n_i$. $\square$

**Lemma B.7.** *Suppose $\mathrm{cost}(\hat{\mathcal{C}}; \mathcal{X}) \leq \alpha \Delta_k^2(\mathcal{X})$ for some $\alpha < \frac{1 - 802\phi^2}{800}$. Then there exists a permutation $\pi : [k] \to [k]$ such that $d(\hat{\mathcal{S}}_i, \mathcal{S}_{\pi(i)}^*) \leq D_i/10$ for every $i = 1, \cdots, k$.*

*Proof.* Pick $\rho = \frac{800\phi^2}{1-2\phi^2}$. By conditions on $\alpha$ we have $\alpha \leq (\frac{1}{\rho} - 1)\phi^2$. In the remainder of the proof we show that for every $i \in [k]$, there exists some $j \in [k]$ such that $d(\mathcal{S}_i^*, \hat{\mathcal{S}}_j) \leq \frac{2d_i}{\sqrt{\rho}} \leq D_i/10$, where the last inequality is due to Proposition B.5 and the choice of $\rho$. This is sufficient for the conclusion

in Lemma B.7 since no two subspaces $\mathcal{S}_i^*$ and $\mathcal{S}_{i'}^*$ can be within the range of $D_i/10$ to the same subspace $\hat{\mathcal{S}}_j$ due to the definition of $D_i$ and triangle inequality presented in Proposition A.1.

Assume by way of contradiction that there exists $i \in [k]$ such that $d(\mathcal{S}_i^*, \hat{\mathcal{S}}_j) > \frac{2d_i}{\sqrt{\rho}}$. This implies that any point in $\mathcal{X}_i^{\mathrm{cor}} = \{\boldsymbol{x} \in \mathcal{X}_i : d(\boldsymbol{x}, \mathcal{S}_i^*) \leq \frac{r_i}{\sqrt{\rho}}\}$ is at least $\frac{d_i}{\sqrt{\rho}}$ away from any subspace in $\hat{\mathcal{C}}$, due to $d_i \geq r_i$ and the triangle inequality. Therefore, $\mathrm{cost}(\hat{\mathcal{C}}; \mathcal{X}) \geq \frac{|\mathcal{X}_i^{\mathrm{cor}}|}{n} \frac{d_i^2}{\rho} \geq (\frac{1}{\rho} - 1) \frac{n_i}{n} d_i^2$, where the last inequality is due to Proposition B.6. Finally, $(\frac{1}{\rho} - 1) \frac{n_i}{n} d_i^2 = (\frac{1}{\rho} - 1) \frac{n_i}{n} \cdot \phi^2 n \Delta_{k-1}^2(\mathcal{X}) > \alpha n_i \Delta_{k-1}^2(\mathcal{X}) \geq \alpha \Delta_{k-1}^2(\mathcal{X})$, and hence the contradiction. $\qquad\square$

**Lemma B.8.** *Fix a candidate subspace set $\hat{\mathcal{C}} = \{\hat{\mathcal{S}}_1, \cdots, \hat{\mathcal{S}}_2\}$. Define $\hat{\mathcal{R}}_i = \{\boldsymbol{x} \in \mathcal{X} : d(\boldsymbol{x}, \hat{\mathcal{S}}_i) \leq d(\boldsymbol{x}, \hat{\mathcal{S}}_j) + \hat{D}_i/4, \forall j\}$, where $\hat{D}_i = \min_{j \neq i} d(\hat{\mathcal{S}}_i, \hat{\mathcal{S}}_j)$. Suppose there exists a permutation $\pi : [k] \to [k]$ such that $d(\hat{\mathcal{S}}_i, \mathcal{S}_{\pi(i)}^*) \leq D_{\pi(i)}/10$ for every $i$, where $D_i = \min_{j \neq i} d(\mathcal{S}_i^*, \mathcal{S}_j^*)$. Then we have $\mathcal{X}_{\pi(i)} \subseteq \hat{\mathcal{R}}_i$ and further more $|\mathcal{X}_{\pi(i)}| \geq \beta |\hat{\mathcal{R}}_i|$ for $\beta = \frac{1 - 2\phi^2}{1 + 48\phi^2}$.*

*Proof.* Without loss of generality we assume $\pi(i) = i$; that is, $d(\hat{\mathcal{S}}_i, \mathcal{S}_i^*) \leq D_i/10$ for every $i = 1, \cdots, k$. By triangle inequality in Proposition A.1, we have $\frac{4}{5} D_i \leq \hat{D}_i \leq \frac{6}{5} D_i$. Fix an arbitrary $\boldsymbol{x} \in \mathcal{X}_i$. By definition, $d(\boldsymbol{x}, \mathcal{S}_i^*) \leq d(\boldsymbol{x}, \mathcal{S}_j^*)$. Therefore, $d(\boldsymbol{x}, \hat{\mathcal{S}}_i) \leq d(\boldsymbol{x}, \mathcal{S}_i^*) + \frac{D_i}{10} \leq d(\boldsymbol{x}, \mathcal{S}_j^*) + \frac{D_i}{10} \leq d(\boldsymbol{x}, \hat{\mathcal{S}}_j) + \frac{D_i}{5} \leq d(\boldsymbol{x}, \hat{\mathcal{S}}_j) + \frac{\hat{D}_i}{4}$. Therefore, $\mathcal{X}_i \subseteq \hat{\mathcal{R}}_i$.

We next prove that $|\mathcal{X}_i| \geq \beta |\hat{\mathcal{R}}_i|$. The approach we take is to assume $|\mathcal{X}_i| = \beta |\hat{\mathcal{R}}_i|$ for some real number $\beta$ and show that $\beta \geq \frac{1 - 2\phi^2}{1 + 48\phi^2}$. Let $a_j = \frac{|\hat{\mathcal{R}}_i \cap \mathcal{X}_j|}{|\hat{\mathcal{R}}_i|}$ and we arbitrarily assign $\frac{a_j n_i}{1 - a_i}$ points in $\mathcal{X}_i$ to the cluster associated with subspace $\mathcal{S}_j^*$. This will clear the $\mathcal{S}_i^*$ subspace since $\sum_{j \neq i} \frac{a_j n_i}{1 - a_i} = n_i$. As a result, we have

$$
\begin{aligned}
n \Delta_{k-1}^2(\mathcal{X}) &\leq n \Delta_k^2(\mathcal{X}) - n_i \Delta_1(\mathcal{X}_i) + \sum_{\boldsymbol{x} \in \mathcal{X}_i} d^2(\boldsymbol{x}, \mathcal{S}_j^*) \\
&\leq n \Delta_k^2(\mathcal{X}) + n_i \Delta_1(\mathcal{X}_i) + 2 \sum_{j \neq i} \frac{a_j n_i}{1 - a_i} \cdot d^2(\mathcal{S}_i^*, \mathcal{S}_j^*) \\
&\leq 2n \Delta_k^2(\mathcal{X}) + \frac{2\beta}{1 - \beta} \sum_{j \neq i} a_j |\hat{\mathcal{R}}_i| d^2(\mathcal{S}_i^*, \mathcal{S}_j^*).
\end{aligned}
$$

The last inequality is due to the fact that $\frac{n_i}{a_i} = \frac{|\mathcal{X}_i| \cdot |\hat{\mathcal{R}}_i|}{|\mathcal{X}_i \cap \hat{\mathcal{R}}_i|} = |\hat{\mathcal{R}}_i|$ and $\frac{a_i}{1 - a_i} = \frac{|\mathcal{X}_i|}{|\hat{\mathcal{R}}_i| - |\mathcal{X}_i|} \leq \frac{\beta}{1 - \beta}$.

On the other hand, for any $\boldsymbol{y} \in \mathcal{X}_j \cap \hat{\mathcal{R}}_i$, one has $d(\boldsymbol{y}, \mathcal{S}_i^*) \leq d(\boldsymbol{y}, \hat{\mathcal{S}}_i) + \frac{D_i}{10} \leq d(\boldsymbol{y}, \hat{\mathcal{S}}_j) + \frac{D_i}{5} + \frac{D_i}{10} \leq d(\boldsymbol{y}, \mathcal{S}_j^*) + \frac{3}{10}(D_i + D_j)$. Consequently, $d(\mathcal{S}_i^*, \mathcal{S}_j^*) \leq d(\boldsymbol{y}, \mathcal{S}_i^*) + d(\boldsymbol{y}, \mathcal{S}_j^*) \leq 2d(\boldsymbol{y}, \mathcal{S}_j^*) + \frac{3}{10}(D_i + D_j) \leq 2d(\boldsymbol{y}, \mathcal{S}_j^*) + \frac{3}{5} d(\mathcal{S}_i^*, \mathcal{S}_j^*)$ and hence $d(\mathcal{S}_i^*, \mathcal{S}_j^*) \leq 5 d(\boldsymbol{y}, \mathcal{S}_j^*)$. Subsequently,

$$
\begin{aligned}
n \Delta_{k-1}^2(\mathcal{X}) &\leq 2n \Delta_k^2(\mathcal{X}) + \frac{2\beta}{1 - \beta} \sum_{j \neq i} a_j |\hat{\mathcal{R}}_i| d^2(\mathcal{S}_i^*, \mathcal{S}_j^*) \\
&= 2n \Delta_k^2(\mathcal{X}) + \frac{2\beta}{1 - \beta} \sum_{j \neq i} |\hat{\mathcal{R}}_i \cap \mathcal{X}_j| d^2(\mathcal{S}_i^*, \mathcal{S}_j^*) \\
&\leq 2n \Delta_k^2(\mathcal{X}) + \frac{50\beta}{1 - \beta} \sum_{j \neq i} \sum_{\boldsymbol{y} \in \hat{\mathcal{R}}_i \cap \mathcal{X}_j} d^2(\boldsymbol{y}, \mathcal{S}_j^*) \\
&\leq 2n \Delta_k^2(\mathcal{X}) + \frac{50\beta}{1 - \beta} \cdot n \Delta_k^2(\mathcal{X}).
\end{aligned}
$$

By the well-separatedness of $\mathcal{X}$, we have

$$
\Delta_k^2(\mathcal{X}) \leq \phi^2 \Delta_{k-1}^2(\mathcal{X}) \leq \left(2 + \frac{50\beta}{1 - \beta}\right) \phi^2 \Delta_k^2(\mathcal{X}).
$$

Therefore, $2 + \frac{50\beta}{1 - \beta} \geq 1/\phi^2$, which implies $\beta \geq \frac{1 - 2\phi^2}{1 + 48\phi^2}$. $\qquad\square$

**Lemma B.9.** *Assume $\phi \leq 1/2$. Fix a candidate subspace set $\hat{\mathcal{C}} = \{\hat{\mathcal{S}}_1, \cdots, \hat{\mathcal{S}}_n\}$. Let $\hat{\mathcal{X}}_i = \{\boldsymbol{x} \in \mathcal{X} : d(\boldsymbol{x}, \hat{\mathcal{S}}_i) \leq d(\boldsymbol{x}, \hat{\mathcal{S}}_j), \forall j\}$ denote the set of all data points that are clustered into $\hat{\mathcal{S}}_i$. Suppose $d(\hat{\mathcal{S}}_i, \mathcal{S}_i^*) \leq D_i/10$ for every $i$. Then $|\hat{\mathcal{X}}_i \triangle \mathcal{X}_i| \leq 150\phi^2 |\mathcal{X}_i|$, where $\triangle$ denotes the symmetric difference operator between two sets.*

*Proof.* We first derive a lower bound on $|\hat{\mathcal{X}}_i \cap \mathcal{X}_i|$. We first claim that for any $\boldsymbol{x} \in \mathcal{X}$, $d(\boldsymbol{x}, \mathcal{S}_i^*) \leq \frac{2}{5}D_i$ yields $\boldsymbol{x} \in \hat{\mathcal{X}}_i$. To see this, note that $d(\boldsymbol{x}, \hat{\mathcal{S}}_i) \leq d(\boldsymbol{x}, \mathcal{S}_i^*) + d(\mathcal{S}_i^*, \hat{\mathcal{S}}_i) \leq \frac{2}{5}D_i + \frac{1}{10}D_i = \frac{1}{2}D_i$ and for every $j \neq i$, $d(\boldsymbol{x}, \hat{\mathcal{S}}_j) \geq d(\mathcal{S}_i^*, \hat{\mathcal{S}}_j) - d(\boldsymbol{x}, \mathcal{S}_i^*) \geq \frac{9}{10}D_i - \frac{2}{5}D_i = \frac{1}{2}D_i$. Therefore, $d(\boldsymbol{x}, \hat{\mathcal{S}}_i) \leq d(\boldsymbol{x}, \hat{\mathcal{S}}_j)$ for every $j \neq i$.

On the other hand, by Proposition B.5 $d(\boldsymbol{x}, \mathcal{S}_i^*) \leq \frac{r_i}{\sqrt{\rho'}}$ with $\rho' = \frac{25\phi^2}{2(1-2\phi^2)}$ implies $d(\boldsymbol{x}, \mathcal{S}_i^*) \leq \frac{2}{5}D_i$. Consequently, by Proposition B.6 we have $|\hat{\mathcal{X}}_i \cap \mathcal{X}_i| \geq |\{\boldsymbol{x} \in \mathcal{X}_i : d(\boldsymbol{x}, \mathcal{S}_i^*) \leq \frac{r_i}{\sqrt{\rho'}}| \geq (1-\rho')|\mathcal{X}_i|$. In addition, Lemma B.8 asserts that $|\mathcal{X}_i| \geq \beta|\hat{\mathcal{R}}_i| \geq \beta|\mathcal{X}_i|$. Therefore, $|\hat{\mathcal{X}}_i \triangle \mathcal{X}_i| \leq (2\rho' + \frac{1}{\beta} - 1)|\mathcal{X}_i| \leq \frac{75\phi^2}{1-2\phi^2}|\mathcal{X}_i| \leq 150\phi^2 \mathcal{X}_i$, assuming $\phi \leq \frac{1}{2}$. $\square$

**Proposition B.10.** *Let $\sigma_q^2(\mathbf{X}_i)$ denote the qth largest singular value of $\mathbf{X}_i$. We then have $\sigma_q^2(\mathbf{X}_i) \geq n\psi$ and $\sigma_{q+1}^2(\mathbf{X}_i) \leq n\eta$.*

*Proof.* By principal component analysis, $n\Delta_k^2(\mathcal{X}) = \sum_{i=1}^{k} \sum_{p \geq q+1} \sigma_p^2(\mathcal{X}_i)$. Therefore, $n\Delta_{k,-1}^2(\mathcal{X}) \leq n\Delta_k^2(\mathcal{X}) + \sigma_q^2(\mathcal{X}_i)$ for any $i$. Since $\mathcal{X}$ is $(\phi, \eta, \psi)$-well separated, we have $n\Delta_{k,-1}^2 \geq n\Delta_k^2 + n\psi$. Consequently, $\sigma_q^2(\mathcal{X}_i) \geq n\psi$. On the other hand, we have $n\Delta_{k,+1}^2(\mathcal{X}) \leq n\Delta_k^2(\mathcal{X}) - \sigma_{q+1}^2(\mathcal{X}_i)$ for any $i$ and $n\Delta_{k,+1}^2(\mathcal{X}) \geq n\Delta_k^2 - n\eta$. Hence $\sigma_{q+1}^2(\mathcal{X}_i) \leq n\eta$. $\square$

**Lemma B.11.** *Following the same notations in Lemma B.11. Suppose $\phi^2 < 1/150$. If $|\hat{\mathcal{X}}_i \triangle \mathcal{X}_i| \leq 150\phi^2 |\mathcal{X}_i|$ holds for every $i$, then $d(\mathcal{S}_i^*, \hat{\mathcal{S}}_i) \leq \frac{600\sqrt{2}\phi^2}{(1-150\phi^2)(\psi-\eta)}$.*

*Proof.* Let $\mathcal{B}_i = \mathcal{X}_i \cap \hat{\mathcal{X}}_i$, $\mathcal{Y}_i = \mathcal{X}_i \backslash \mathcal{B}_i$ and $\mathcal{Z}_i = \hat{\mathcal{X}}_i \backslash \mathcal{B}_i$. Since $|\hat{\mathcal{X}}_i \triangle \mathcal{X}_i| \leq 150\phi^2 |\mathcal{X}_i|$, we have $|\mathcal{B}_i| \geq (1-150\phi^2)|\mathcal{X}_i|$ and $|\mathcal{Y}_i|, |\mathcal{Z}_i| \leq 150\phi^2 |\mathcal{X}_i|$. Let $\mathbf{B}_i, \mathbf{Y}_i, \mathbf{Z}_i$ be the matrices associated with $\mathcal{B}_i, \mathcal{Y}_i$ and $\mathcal{Z}_i$. Define $\mathbf{A}_i = \frac{\mathbf{B}_i \mathbf{B}_i^\top + \mathbf{Y}_i \mathbf{Y}_i^\top}{|\mathcal{B}_i| + |\mathcal{Y}_i|}$ and $\widetilde{\mathbf{A}}_i = \frac{\mathbf{B}_i \mathbf{B}_i^\top + \mathbf{Z}_i \mathbf{Z}_i^\top}{|\mathcal{B}_i| + |\mathcal{Z}_i|}$. By principal component analysis, $\mathcal{S}_i^*$ and $\hat{\mathcal{S}}_i$ are the span of top-$q$ eigenvectors associated with $\mathbf{A}_i$ and $\widetilde{\mathbf{A}}_i$. By Wedin's Theorem (Theorem F.2 in Appendix F), the distance $d(\mathcal{S}_i^*, \hat{\mathcal{S}}_i)$ can be bounded by upper bounding the perturbation between $\mathbf{A}_i$ and $\widetilde{\mathbf{A}}_i$, for example, $\|\mathbf{A}_i - \widetilde{\mathbf{A}}_i\|_F$.

Define $\bar{\mathbf{A}}_i = \frac{\mathbf{B}_i \mathbf{B}_i^\top}{|\mathcal{B}_i|}$ and consider separately $\|\mathbf{A}_i - \bar{\mathbf{A}}_i\|_F$ and $\|\widetilde{\mathbf{A}}_i - \bar{\mathbf{A}}_i\|_F$. By definition, we have

$$
\begin{aligned}
\|\mathbf{A}_i - \bar{\mathbf{A}}_i\|_F &= \left\| \frac{\mathbf{B}_i \mathbf{B}_i^\top}{|\mathcal{B}_i|} - \frac{\mathbf{B}_i \mathbf{B}_i^\top + \mathbf{Y}_i \mathbf{Y}_i^\top}{|\mathcal{B}_i| + |\mathcal{Y}_i|} \right\|_F \\
&\leq \left\| \left( \frac{1}{|\mathcal{B}_i|} - \frac{1}{|\mathcal{B}_i| + |\mathcal{Y}_i|} \right) \mathbf{B}_i \mathbf{B}_i^\top \right\|_F + \left\| \frac{\mathbf{Y}_i \mathbf{Y}_i^\top}{|\mathcal{B}_i| + |\mathcal{Y}_i|} \right\|_F \\
&\leq \frac{|\mathcal{Y}_i| \cdot \|\mathbf{B}_i\|_F^2}{|\mathcal{B}_i|(|\mathcal{B}_i| + |\mathcal{Y}_i|)} + \frac{\|\mathbf{Y}_i\|_F^2}{|\mathcal{B}_i| + |\mathcal{Y}_i|} \\
&\leq \frac{|\mathcal{Y}_i| \cdot |\mathcal{B}_i|}{|\mathcal{B}_i|(|\mathcal{B}_i| + |\mathcal{Y}_i|)} + \frac{|\mathcal{Y}_i|}{|\mathcal{B}_i| + |\mathcal{Y}_i|} \\
&= \frac{2|\mathcal{Y}_i|}{|\mathcal{B}_i| + |\mathcal{Y}_i|} \leq \frac{300\phi^2}{1 - 150\phi^2}.
\end{aligned}
$$

Using essentially the same line of argument one can show $\|\widetilde{\mathbf{A}}_i - \bar{\mathbf{A}}_i\|_F \leq \frac{300\phi^2}{1-150\phi^2}$ as well. Therefore, $\|\mathbf{A}_i - \widetilde{\mathbf{A}}_i\|_F \leq \frac{600\phi^2}{1-150\phi^2}$. Applying Wedin's Theorem and Proposition B.10 we get

$$
d(\mathcal{S}_i^*, \hat{\mathcal{S}}_i) \leq \frac{\sqrt{2}\|\mathbf{A}_i - \widetilde{\mathbf{A}}_i\|_F}{\sigma_q(\mathbf{A}_i) - \sigma_{q+1}(\mathbf{A}_i)} \leq \frac{\sqrt{2}|\mathcal{X}_i|\|\mathbf{A}_i - \widetilde{\mathbf{A}}_i\|_F}{\sigma_q^2(\mathcal{X}_i) - \sigma_{q+1}^2(\mathcal{X}_i)} \leq \frac{\sqrt{2}|\mathcal{X}_i|\|\mathbf{A}_i - \widetilde{\mathbf{A}}_i\|_F}{n(\psi - \eta)} \leq \frac{600\sqrt{2}\phi^2}{(1 - 150\phi^2)(\psi - \eta)}.
$$

□

Combining Lemma B.7 to B.11 we arrive at a proof of the key lemma.

*Proof of Lemma 3.3.* Given $a < \frac{1-802\phi^2}{800\phi^2}$ and $\mathcal{X}$ being $(\phi, \eta, \psi)$-well separated, we have $\text{cost}(\hat{\mathcal{C}}; \mathcal{X}) \leq \alpha\Delta_{k-1}^2(\mathcal{X})$ for some $\alpha < \frac{1-802\phi^2}{800}$. By Lemma B.7, $d(\mathcal{S}_i^*, \hat{\mathcal{S}}_i) \leq D_i/10$ hold for every $i = 1, \cdots, k$, after possible rearrangement of $\{\hat{\mathcal{S}}_i\}_{i=1}^k$. Applying Lemma B.8, B.9 and B.11 we obtain $d(\mathcal{S}_i^*, \hat{\mathcal{S}}_i) \leq \frac{600\sqrt{2}\phi^2}{(1-150\phi^2)(\psi-\eta)}$. Finally, $d_W(\hat{\mathcal{C}}, \mathcal{C}^*) \leq \sqrt{k} \max_i d(\mathcal{S}_i^*, \hat{\mathcal{S}}_i) \leq \frac{600\sqrt{2}\phi^2\sqrt{k}}{(1-150\phi^2)(\psi-\eta)}$. □

## Appendix C  Proofs of sample-aggregate private subspace clustering: the stochastic case

In this section we prove Theorem 3.6 that details a stability result for threshold-based subspace clustering under the stochastic datasetting. We first cite the following lemma from [14] which states that (under certain separation conditions) with high probability the similarity graph $G$ recovered by the robust TSC algorithm has no false connections; that is, two data points $i$ and $j$ are connected in $G$ only if they belong to the same cluster (subspace).

**Lemma C.1** ([14], Theorem 3; no false connection of TSC). *Suppose $s \leq \min n_\ell/6$ and*

$$\sqrt{1 - \min_{\ell \neq \ell'} \frac{d^2(\mathcal{S}_\ell^*, \mathcal{S}_{\ell'}^*)}{q}} + \frac{\sigma(1+\sigma)}{\sqrt{\log n}}\frac{\sqrt{q}}{\sqrt{d}} \leq \frac{1}{15\log n} \tag{14}$$

*with $d \geq 6\log n$; then the similarity graph $G$ constructed by Algorithm 2 has no false connections with probability at least $1 - \frac{10}{n} - \sum_\ell n_\ell e^{-c(n_\ell-1)}$ for some absolute constant $c > 0$.*

Based on Lemma C.1, to prove Lemma 3.5 it remains to show that data points within the same cluster are indeed *connected* in the similarity graph $G$. The proof is presented in Appendix C.1. With Lemma C.1 and 3.5, Theorem 3.6 can be easily proved as follows:

*Proof of Theorem 3.6.* By Lemma C.1 and 3.5, we know that under the stated conditions the similarity graph $G$ output by the TSC algorithm has no false connections and is connected per cluster. Fix a cluster $\ell$ and consider the observed data points $\mathbf{X}^{(\ell)} = \mathbf{Y}^{(\ell)} + \mathbf{E}^{(\ell)}$. Since both the signal $\mathbf{Y}^{(\ell)}$ and the noise $\mathbf{E}^{(\ell)}$ are stochastic, by standard analysis of PCA one can show that the top-$q$ subspace of $\mathbf{X}^{(\ell)}$ converges to the underlying subspace $\mathcal{S}_\ell^*$ in probability as the number of data points $n_\ell$ goes to infinity [43]. The theorem then holds because $m = o(n)$ and hence $n' = n/m \to \infty$. □

### C.1  Proof of Lemma 3.5

**Proposition C.2.** *Suppose $\mathbf{y}_i = \mathbf{x}_i + \boldsymbol{\varepsilon}_i$ with $\boldsymbol{\varepsilon}_i \sim \mathcal{N}(\mathbf{0}, \frac{\sigma^2}{d}\mathbf{I}_d)$ and $\sigma > 0$. Then with probability at least $1 - n^2 e^{-\sqrt{d}} - 2/n$ the following holds:*

$$\left|\langle \mathbf{y}_i, \mathbf{y}_j \rangle - \langle \mathbf{x}_i, \mathbf{x}_j \rangle\right| \leq (2\sqrt{5}\sigma + 5\sigma^2)\sqrt{\frac{6\log n}{d}}; \quad \forall i, j \in \{1, \cdots, n\}, i \neq j. \tag{15}$$

*Proof.* Applying Theorem F.3 (in Appendix F) and set $t = 4$ and $\rho = \sqrt{6\log n/d}$ in Theorem F.3. Applying also the union bound over all $(i, j) \in \{1, \cdots, n\}$ pairs with $i \neq j$. Then the following

holds uniformly for all $i \neq j$ with probability at least $1 - n^2 e^{-d} - 2/n$:

$$\|\boldsymbol{\varepsilon}_i\|_2 \leq \sqrt{5}\sigma;$$

$$\left|\langle \boldsymbol{\varepsilon}_i, \boldsymbol{x}_j \rangle\right| \leq \sqrt{5}\sigma \cdot \sqrt{\frac{6 \log n}{d}}, \quad \forall i, j;$$

$$\left|\langle \boldsymbol{\varepsilon}_i, \boldsymbol{\varepsilon}_j \rangle\right| \leq 5\sigma^2 \cdot \sqrt{\frac{6 \log n}{d}}, \quad \forall i \neq j.$$

The proof is then completed by noting that

$$
\begin{aligned}
\left|\langle \boldsymbol{y}_i, \boldsymbol{y}_j \rangle - \langle \boldsymbol{x}_i, \boldsymbol{x}_j \rangle\right| &\leq \left|\langle \boldsymbol{\varepsilon}_i, \boldsymbol{x}_j \rangle\right| + \left|\langle \boldsymbol{\varepsilon}_j, \boldsymbol{x}_i \rangle\right| + \left|\langle \boldsymbol{\varepsilon}_i, \boldsymbol{\varepsilon}_j \rangle\right| \\
&\leq (2\sqrt{5}\sigma + 5\sigma^2)\sqrt{\frac{6 \log n}{d}}.
\end{aligned}
$$

$\square$

**Lemma C.3** ([14], Lemma 3; extracted from the proof of Lemma 6.2. in [42]). *Let* $S^{d-1} = \{\boldsymbol{x} \in \mathbb{R}^d : \|\boldsymbol{x}\|_2 = 1\}$ *denote the d-dimensional unit sphere. For an arbitrary* $\boldsymbol{p} \in S^{d-1}$, *define* $C(\boldsymbol{p}, \theta) = \{\boldsymbol{q} \in S^{d-1} : \vartheta(\boldsymbol{p}, \boldsymbol{q}) \leq \theta\}$ *where* $\vartheta(\boldsymbol{p}, \boldsymbol{q}) = \arccos(\langle \boldsymbol{p}, \boldsymbol{q} \rangle)$ *is the angle between* $\boldsymbol{p}$ *and* $\boldsymbol{q}$. *Let* $\mathcal{L}(\cdot)$ *denote the Lebesgue area of a region and* $\Theta(\cdot)$ *be the inverse function of* $\mathcal{L}(C(\boldsymbol{p}, \theta))$ *with respect to* $\theta$. *Then for each* $d \geq 1$ *and* $M \geq 1$, *there exists a partition* $R_1, \cdots, R_M$ *of the unit sphere* $\mathbb{S}^{d-1}$ *such that [5]* $\sup_{\boldsymbol{x}, \boldsymbol{y} \in R_m} \vartheta(\boldsymbol{x}, \boldsymbol{y}) \leq \theta^*$ *for every* $m = 1, \cdots, M$. *Here* $\theta^* = 8\Theta(\mathcal{L}(S^{d-1})/M)$.

We are now ready to prove Lemma 3.5.

*Proof of Lemma 3.5.* By Lemma C.1 we already know that the similarity graph $G$ has no false connections. Fix a cluster $\ell \in \{1, \cdots, k\}$. Let $\boldsymbol{x}_i^{(\ell)} = \mathbf{U}^{(\ell)} \boldsymbol{a}_i^{(\ell)}$ where $\boldsymbol{a}_i^{(\ell)} \in S^{q-1}$. Set $M = n_\ell/(\gamma \log n_\ell)$ and let $R_1, \cdots, R_M$ be a partition of the unit sphere $S^{q-1}$ as characterized in Lemma C.3. Here $q$ is the intrinsic rank of an underlying subspace. We need to prove the following two properties hold with high probability: (A) every region $R_m$ contains at least one point in $\mathcal{A}^{(\ell)} = \{\boldsymbol{a}_i^{(\ell)}\}_{i=1}^{n_\ell}$; (B) for every $\boldsymbol{a}_i^{(\ell)}$, all data points $\boldsymbol{a}_j^{(\ell)}$ belonging to the neighboring region of the region containing $\boldsymbol{a}_i^{(\ell)}$ are connected with $\boldsymbol{a}_i^{(\ell)}$.

Property (A) is easy to prove. By union bound, the probability that some region is empty can be upper bounded by

$$M\left(1 - \frac{1}{M}\right)^{n_\ell} \leq M e^{-n_\ell/M} = \frac{n_\ell^{1-\gamma}}{\gamma \log n_\ell}.$$

We next turn to prove Property (B). Unlike the noiseless case, the $s$-nearest-neighbor graph is computed based on the noise-perturbed data points $\{\boldsymbol{y}_i\}_{i=1}^n$. It is no longer true that data points belonging to neighboring regions have larger inner products compared to data points that do not belong to the same or neighboring regions. Hence, we adopt a different argument from the one presented in [14]. Instead of showing that $|C(\boldsymbol{x}_i^{(\ell)}, 3\theta^*)| \leq \tilde{s} = s/2$, we show that $|C(\boldsymbol{a}_i^{(\ell)}, r\theta^*)| \leq \tilde{s}$ for some $r \gg 3$ and in addition $|\langle \boldsymbol{y}_j^{(\ell)}, \boldsymbol{y}_i^{(\ell)} \rangle| > |\langle \boldsymbol{y}_{j'}^{(\ell)}, \boldsymbol{y}_i^{(\ell)} \rangle|$ for every $\boldsymbol{a}_j^{(\ell)} \notin C(\boldsymbol{a}_i^{(\ell)}, r\theta^*)$ and $\boldsymbol{a}_{j'}^{(\ell)} \in C(\boldsymbol{x}_i^{(\ell)}, 3\theta^*)$. [6] This guarantees that all points in $C(\boldsymbol{a}_i^{(\ell)}, 3\theta^*)$ are connected to $\boldsymbol{y}_i^{(\ell)}$ in the $s$-nearest-neighbor graph.

Fix $\boldsymbol{a}_i^{(\ell)} \in \mathcal{A}^{(\ell)}$ and set $r$ such that

$$r\theta^* = \left(\frac{0.01(q/2 - 1)(q - 1)}{\sqrt{\pi}}\right)^{\frac{1}{q-1}}. \tag{16}$$

Define $p = \mathcal{L}(C(\boldsymbol{a}_i^{(\ell)}, r\theta^*))/\mathcal{L}(S^{q-1})$, where $\theta^*$ is given in Lemma C.3. By definition, $\mathbb{E}[|C(\boldsymbol{a}_i^{(\ell)}, r\theta^*)|] = pn_\ell$. Note that by symmetry $p$ does not depend on $\boldsymbol{a}_i^{(\ell)}$. Set $\bar{s} = (0.01 + p)n_\ell$. We then have

$$\frac{\bar{s}}{n_\ell} = 0.01 + \frac{\mathcal{L}(C(\boldsymbol{a}_i^{(\ell)}, r\theta^*))}{L(S^{q-1})} \le 0.01 + \frac{\mathcal{L}(S^{q-2})(r\theta^*)^{q-1}}{L(S^{q-1})(q-1)} \le 0.01 + \frac{\sqrt{\pi}\Gamma(\frac{q-1}{2})(r\theta^*)^{q-1}}{\Gamma(\frac{q}{2})(q-1)} \le 0.02. \tag{17}$$

The second inequality is an application of Eq. (5.2) in [42] and the last inequality is due to Eq. (16). On the other hand, by tail bounds of binomial distribution (Theorem 1 in [41]) we have

$$\Pr\left[\left|C(\boldsymbol{a}_i^{(\ell)}, r\theta^*)\right| > n_\ell(p + 0.01)\right] \le e^{-\frac{0.01^2 n_\ell^2}{2(pn_\ell + 0.01 n_\ell/3)}} \le e^{-\frac{n_\ell}{400}}, \tag{18}$$

where in the last inequality we used the fact that $pn_\ell \le 0.01n_\ell$. Since $\bar{s} \le 0.02n_\ell \le \tilde{s}$, we proved that with probability at least $1 - e^{-\frac{n_\ell}{400}}$ there will be no more than $\tilde{s}$ data points contained in $C(\boldsymbol{a}_i^{(\ell)}, r\theta^*)$.

The final step of the proof is to show that $|\langle \boldsymbol{y}_j^{(\ell)}, \boldsymbol{y}_i^{(\ell)}\rangle| > |\langle \boldsymbol{y}_{j'}^{(\ell)}, \boldsymbol{y}_j^{(\ell)}\rangle|$ for every $\boldsymbol{a}_j^{(\ell)} \notin C(\boldsymbol{a}_i^{(\ell)}, r\theta^*)$ and $\boldsymbol{a}_{j'}^{(\ell)} \in C(\boldsymbol{x}_i^{(\ell)}, 3\theta^*)$. By Proposition C.2, we have with probability at least $1 - ne^{-\sqrt{d}}$

$$\left|\langle \boldsymbol{y}_j^{(\ell)}, \boldsymbol{y}_i^{(\ell)}\rangle\right| \le \left|\langle \boldsymbol{a}_j^{(\ell)}, \boldsymbol{a}_i^{(\ell)}\rangle\right| + (2\sqrt{5}\sigma + 5\sigma^2)\sqrt{\frac{6\log n}{d}} \le \cos(r\theta^*) + (2\sqrt{5}\sigma + 5\sigma^2)\sqrt{\frac{6\log n}{d}} \tag{19}$$

and

$$\left|\langle \boldsymbol{y}_{j'}^{(\ell)}, \boldsymbol{y}_i^{(\ell)}\rangle\right| \ge \left|\langle \boldsymbol{a}_{j'}^{(\ell)}, \boldsymbol{a}_i^{(\ell)}\rangle\right| - (2\sqrt{5}\sigma + 5\sigma^2)\sqrt{\frac{6\log n}{d}} \ge \cos(3\theta^*) - (2\sqrt{5}\sigma + 5\sigma^2)\sqrt{\frac{6\log n}{d}}. \tag{20}$$

Since $r\theta^*$ is dictated in Eq. (16), we only need to obtain an upper bound on $\theta^*$. Following the same argument on page 25 in [14] we have

$$\theta^* \le 4\pi\left(\frac{\sqrt{2\pi q}}{M}\right)^{\frac{1}{q-1}} = 4\pi\left(\frac{\gamma\sqrt{2\pi q}\log n_\ell}{n_\ell}\right)^{\frac{1}{q-1}}. \tag{21}$$

Consequently, $|\langle \boldsymbol{y}_j^{(\ell)}, \boldsymbol{y}_i^{(\ell)}\rangle| > |\langle \boldsymbol{y}_{j'}^{(\ell)}, \boldsymbol{y}_j^{(\ell)}\rangle|$ when $\bar{\sigma} = 2\sqrt{5}\sigma + 5\sigma^2$ satisfies

$$\bar{\sigma} < \sqrt{\frac{d}{24\log n}}\left[\cos\left(12\pi\left(\frac{\gamma\sqrt{2\pi q}\log n_\ell}{n_\ell}\right)^{\frac{1}{q-1}}\right) - \cos\left(\left(\frac{0.01(q/2-1)(q-1)}{\sqrt{\pi}}\right)^{\frac{1}{q-1}}\right)\right]. \tag{22}$$

The right-hand side of the above condition is strictly positive if $n_\ell$ satisfies

$$n_\ell > \frac{\gamma\pi\sqrt{2q}\log n_\ell}{0.01(q/2-1)(q-1)} \cdot (12\pi)^{q-1}.$$

$\square$

## Appendix D Supplementary materials for private subspace clustering via the exponential mechanism

### D.1 Proof of Proposition 4.1

*Proof.* Define the score function $h(\cdot; \boldsymbol{\theta})$ as $h(\mathcal{X}; \boldsymbol{\theta}) = \sum_{i=1}^n d^2(\boldsymbol{x}_i, \mathcal{S}_{z_i})$. Since $\|\boldsymbol{x}_i\|_2 \le 1$, it is straightforward that $h(\cdot; \boldsymbol{\theta})$ has global sensitivity upper bounded by 1; that is, $\sup_{d(\mathcal{X}, \mathcal{X}')=1}|h(\mathcal{X}; \boldsymbol{\theta}) - h(\mathcal{X}'; \boldsymbol{\theta})| \le 1$ for all $\boldsymbol{\theta}$. Eq. (10) is then a direct application of the exponential mechanism. $\square$

---

**Algorithm 3** Gibbs sampling for matrix Bingham distribution (Eq. (23))

---

1: **Input**: symmetric matrix $\mathbf{A}$, diagonal matrix $\mathbf{B}$, current sample $\mathbf{U}$, dimensions $d$ and $q$.
2: **for** each $r \in \{1, \cdots, q\}$ in random order **do**
3:     Let $\mathbf{U}_{(r)} \in \mathbb{R}^d$ be the $r$th column of $\mathbf{U}$ and $\mathbf{U}_{(-r)}$ be the matrix excluding $\mathbf{U}_{(r)}$.
4:     Let $\mathbf{N}$ be an orthonormal basis of the null space of $\mathbf{U}_{(-r)}$.
5:     Compute $\boldsymbol{z} = \mathbf{N}^\top \mathbf{U}_{(r)}$ and $\widetilde{\mathbf{A}} = \mathbf{B}_{rr} \mathbf{N}^\top \mathbf{A} \mathbf{N}$.
6:     Update $\boldsymbol{z}$ by Gibbs sampling from the vector Bingham distributrion with parameter $\widetilde{\mathbf{A}}$.
7:     Set $\mathbf{U}_{(r)} = \mathbf{N}\boldsymbol{z}$.
8: **end for**
9: **Output**: the updated sample $\mathbf{U}$.

---

---

**Algorithm 4** Gibbs sampling for vector Bingham distribution (Eq. (24))

---

1: **Input**: symmetric matrix $\mathbf{A}$, current sample $\boldsymbol{x}$, dimension $d$.
2: Let $\mathbf{A} = \mathbf{E}\boldsymbol{\Lambda}\mathbf{E}^\top$, $\boldsymbol{\Lambda} = \mathbf{diag}(\boldsymbol{\lambda})$ be the eigen-decomposition of $\mathbf{A}$. Compute $\boldsymbol{y} = \mathbf{E}^\top \boldsymbol{x}$.
3: **for** each $j \in \{1, \cdots, d\}$ in random order **do**
4:     Compute $q_1, \cdots, q_d$ as $y_1^2/(1 - y_i^2), \cdots, y_d^2/(1 - y_i^2)$.
5:     Sample $\theta \in (0, 1)$ from the density $p(\theta) \propto e^{(\lambda_i - \boldsymbol{q}_{-i}^\top \boldsymbol{\lambda}_{-i})\theta} \times \theta^{-1/2}(1 - \theta)^{(d-3)/2}$.
6:     Set $s = +1$ or $-1$ with equal probability.
7:     Set $y_i = s_i \theta^{1/2}$ and for each $j \neq i$ set $y_j^2 = (1 - \theta)q_j$, leaving the sign unchanged.
8: **end for**
9: **Output**: the updated sample $\boldsymbol{x} = \mathbf{E}\boldsymbol{y}$.

---

## D.2 Gibbs sampling for matrix Bingham distribution

In this section we give details of a Gibbs sampler proposed in [16] for sampling from a matrix Bingham distribution. One component in the Gibbs sampler (the rejection sampling step) is slightly modified to make the sampling more efficient.

The objective is to sample from the following matrix-Bingham distribution:

$$p(\mathbf{U}; \mathbf{A}, \mathbf{B}) \propto \exp(\mathrm{tr}(\mathbf{B}\mathbf{U}^\top \mathbf{A}\mathbf{U})), \tag{23}$$

where $\mathbf{U}$ is a $d \times q$ matrix lying on a Stiefel manifold; that is, $\mathbf{U}^\top \mathbf{U} = \mathbf{I}_{q \times q}$. In our problem $\mathbf{A}$ is an unnormalized sample covariance matrix and $\mathbf{B} = \varepsilon \mathbf{I}_{q \times q}$, with $\varepsilon$ the privacy budget. As a simplified case, when $q = 1$ we arrive at a vector version of the Bingham distribution:

$$p(\boldsymbol{x}; \mathbf{A}) \propto \exp(\boldsymbol{x}^\top \mathbf{A}\boldsymbol{x}), \tag{24}$$

with $\boldsymbol{x}$ constrained on the $d$-dimensional sphere $\{\boldsymbol{x} \in \mathbb{R}^d : \|\boldsymbol{x}\|_2 = 1\}$. Gibbs samplers for both Eq. (23) and (24) were proposed in [16] and presented in Algorithm 3 and 4.

In Algorithm 4, step 4 requires sampling from a non-standard 1-dimensional distribution

$$p(x; k, a) \propto x^{-1/2}(1 - x)^k e^{ax} \cdot \mathbf{1}_{0 < x < 1} =: f(x). \tag{25}$$

In [16] a rejection sampling algorithm was proposed to sample $x$ from Eq. (25), with a $\mathrm{Beta}(1/2, 1 + \min(k, \max(k - a, -1/2)))$ envelope distribution. However, such a distribution is highly inefficient when $|a| \gg 0$ for which no Beta distribution serves as a good envelope distribution. To address this problem, we propose two separate rejection sampling schemes for Eq. (25) when $|a| \gg 0$.

**Case 1:** $a \ll 0$    In this case, the mass of the distribution will concentrate on $x \to 0$. We use Gamma distribution $\Gamma(1/2, 1/|a|)$ truncated on $(0, 1)$ as an envelope distribution. That is, $x \sim g(\cdot)$ and $g(\cdot)$ is defined as

$$g(x) = \frac{1}{Z} \cdot x^{-1/2} e^{ax} \cdot \mathbf{1}_{0 < x < 1},$$

with $Z$ a normalization constant. The constant $M = \sup_x f(x)/g(x)$ can be computed as

$$M = Z \cdot \sup_{0 < x < 1} \frac{x^{-1/2}(1 - x)^k e^{ax}}{x^{-1/2} e^{ax}} \leq Z.$$

The step-by-step algorithm is as follows:

1. Sample $x \sim \Gamma(1/2, 1/|a|)$. If $x \geq 1$, throw away $x$ and re-draw the sample.

2. Sample $u \in (0, 1)$ from the uniform distribution over $(0, 1)$.

3. If $u \leq (1 - x)^k$, accept the sample; otherwise reject the sample and try again.

The proposed rejection sampling algorithm is efficient because when $a \ll 0$, the envelope distribution $g$ has very high density over the region near zero; consequently, $(1 - x)^k$ is close to one and hence the acceptance rate is high.

**Case 2:** $a \gg 0$  In this case, the mass of the distribution will concentrate on $x \to 1$. However, we have a singularity at $x = 0$ (i.e., $\lim_{x \to 0} f(x) = \infty$). This makes the sampling particularly difficult as a distribution proportional to $e^{ax}$ will be infinitely off at the region near zero. To circumvent the problem, we propose a mixture distribution as the envelope, which have good approximation property at both regions near 0 and 1.

First define density $h$ as

$$h(x) = \frac{1}{Z} \cdot (1 - x)^k e^{ax} \cdot \mathbf{1}_{0 < x < 1},$$

where $Z = \int_0^1 (1 - x)^k e^a x \mathrm{d}x$ is the normalization constant. Note that samples from $h(\cdot)$ can be obtained by first sampling $z$ from a Gamma distribution $\Gamma(k + 1, 1/a)$ truncated on $(0, 1)$ and then apply transform of variable $x = 1 - z$. The density of the envelope distribution $g(\cdot)$ is then defined as a mixture distribution:

$$g(x) = \frac{1}{Z} \cdot \mathrm{Beta}(x; 1/2, k + 1) + \left(1 - \frac{1}{Z}\right) \cdot h(x).$$

The constant $M = \sup f(x)/g(x)$ can be computed by

$$
\begin{aligned}
M &= \max\left(\sup_{0 < x \leq 1/a} \frac{f(x)}{g(x)}, \sup_{1/a < x < 1} \frac{f(x)}{g(x)}\right) \\
&\leq \max\left(Z \cdot \sup_{0 < x \leq 1/a} \frac{x^{-1/2}(1 - x)^k e^{ax} \cdot B(1/2, k + 1)}{x^{-1/2}(1 - x)^k}, 2 \cdot \sup_{1/a < x < 1} \frac{x^{-1/2}(1 - x)^k e^{ax}}{(1 - x)^k e^{ax}}\right) \\
&\leq Z \cdot \max(2\sqrt{a}, eB(1/2, k + 1)).
\end{aligned}
$$

Here for the second inequality we apply $Z \geq 2$ for reasonably large $a$ and $B(\cdot, \cdot)$ is the Beta function. The normalization constant $Z$ can be approximately computed using numerical integration. However, empirical evidence suggests that $Z$ is huge for large $a$ values (e.g., $Z > 10^{100}$ if $a > k + 500$). Therefore, we could simply take $Z \to \infty$, which simplifies the rejection sampling algorithm as follows:

1. Sample $z \sim \Gamma(k + 1, 1/a)$. If $z \geq 1$ then throw away $z$ and re-draw the sample.

2. Compute $x = 1 - z$, $M = \max(2\sqrt{a}, \exp(1) \cdot B(1/2, k + 1))$.

3. Sampe $u$ from the uniform distribution over $(0, 1)$.

4. If $u < x^{-1/2}/(M(1 + e^{-ax}\mathrm{Beta}(x; 1/2, k + 1)))$, accept the sample; otherwise reject the sample and try again.

The proposed rejection sampling scheme is efficient because when $a \gg 0$ the density $h(\cdot)$ is very skewed to one. Therefore, $x^{-1/2}$ will be close to 1 and $e^{-ax}$ will be very samll, which means the acceptance rate is high.

### D.3  Justification of generative model in Section 4.2

Recall the generative model presented in Section 4.2:

1. For each $\ell \in [k]$, sample $\mathbf{U}_\ell$ (orthonormal basis of $\mathcal{S}_\ell$) uniformly at random from $\mathbb{S}_q^d$.

2. For each $i \in [n]$, sample $z_i \in [k]$ such that $\Pr[z_i = j] = 1/k$, $\boldsymbol{y}_i$ uniformly at random from the $q$-dimensional unit ball, and $\boldsymbol{w}_i \sim \mathcal{N}(0, \mathbf{I}_d/\varepsilon)$. Set $\boldsymbol{x}_i = \mathbf{U}_\ell \boldsymbol{y}_i + \mathcal{P}_{\mathbf{U}_\ell^\perp} \boldsymbol{w}_i$.

---

**Algorithm 5** Differentially private query answering via the SuLQ framework

---

1: **Input**: query parameters $\mathcal{S}_1, \cdots, \mathcal{S}_k \in \mathbb{R}_d^q$, $\ell \in [k]$; privacy parameters $\varepsilon, \delta > 0$.
2: Let $\mathbf{A}_\ell = \{\boldsymbol{x}_i : \text{argmin}_{\ell'} d(\boldsymbol{x}_i, \mathcal{S}_{\ell'}) = \ell\}$ and form $\mathbf{B} = \mathbf{A}_\ell \mathbf{A}_\ell^\top$.
3: **Noise calibration**: Set $\widetilde{\mathbf{B}} = \mathbf{B} + \sigma\mathbf{W}$, where $\mathbf{W}$ is a standard Normal random matrix and $\sigma = 2\sqrt{2\ln(1.25/\delta)}/\varepsilon$.
4: **Singular value decomposition**: Let $\widetilde{\mathbf{B}} = \mathbf{U}\mathbf{V}\mathbf{D}^\top$ be the top-$q$ singular value decomposition of $\widetilde{\mathbf{B}}$. $\mathbf{U} \in \mathbb{R}^{d \times q}$ denotes the top $q$ left singular vectors of $\widetilde{\mathbf{B}}$.
5: **Output**: new subspace $\mathcal{S}'_\ell$ spanned by columns of $\mathbf{U}$.

---

In this section we derive a Gibbs sampler for the considered model and show that the derived Gibbs sampler is identical to the one presented in Section 4.1. This result establishes formal connection between our proposed Gibbs sampling algorithm for private subspace clustering and a probabilistic graphical model that resembles the mixtures of probabilistic PCA (MPPCA, [27]) model.

First we note that the prior distribution specified in the generative model is completely non-informative; that is, $p_0(\boldsymbol{\theta}) = p_0(\boldsymbol{\theta}')$ for any $\boldsymbol{\theta} = (\mathcal{C}, \boldsymbol{x}, \boldsymbol{y}, \boldsymbol{z})$ and $\boldsymbol{\theta}' = (\mathcal{C}', \boldsymbol{x}', \boldsymbol{y}', \boldsymbol{z}')$. On the other hand, the likelihood model is as follows:

$$p(\boldsymbol{x}_i|z_i = \ell, \boldsymbol{y}_i, \mathcal{C}) = \begin{cases} \mathcal{N}(\boldsymbol{x}_i; \mathbf{U}_\ell \boldsymbol{y}_i, \mathbf{I}_d/\varepsilon), & \text{if } \mathcal{P}_{\mathcal{S}_\ell} \boldsymbol{x}_i = \mathbf{U}_\ell \boldsymbol{y}_i; \\ 0, & \text{otherwise.} \end{cases} \tag{26}$$

Here $\mathbf{U}_\ell \in \mathbb{R}^{d \times q}$ is an orthonormal basis associated with $\mathcal{S}_\ell$ and $\mathcal{P}_{\mathcal{S}_\ell}$ stands for the projection operator onto subspace $\mathcal{S}_\ell$. Integrating $\boldsymbol{y}_i$ out we obtain

$$p(\boldsymbol{x}_i|z_i = \ell, \mathcal{C}) \propto \exp\left(-\frac{\varepsilon}{2} \cdot d^2(\boldsymbol{x}_i, \mathcal{S}_\ell)\right). \tag{27}$$

A Gibbs sampler can then be derived as follows:

**Update of $z_i$**  By Eq. (27), the conditional distribution of $z_i$ is

$$p(z_i = \ell|\boldsymbol{x}_i, \mathcal{C}) \propto p_0(z_i = \ell)p(\boldsymbol{x}_i|z_i = \ell, \mathcal{C}) \propto \exp\left(-\frac{\varepsilon}{2} \cdot d^2(\boldsymbol{x}_i, \mathcal{S}_\ell)\right).$$

Therefore, we can sample $z_i$ from a normalized categorical distribution as specified above.

**Update of $\mathcal{S}_\ell$**  By Eq. (27), the conditional distribution of $\mathcal{S}_\ell$ is

$$p(\mathcal{S}_\ell|\boldsymbol{x}, \boldsymbol{z}) \propto p_0(\mathcal{S}_\ell) \prod_{z_i = \ell} p(\boldsymbol{x}_i|z_i = \ell, \mathcal{S}_\ell) \propto \exp\left(-\frac{\varepsilon}{2} \cdot \sum_{z_i = \ell} d^2(\boldsymbol{x}_i, \mathcal{S}_\ell)\right).$$

Denote $\mathbf{A}_\ell = \{\boldsymbol{x}_i : z_i = \ell\}$ as all data points in cluster $\ell$ and let $\mathbf{U}_\ell$ be the orthonormal basis of $\mathcal{S}_\ell$. We then have

$$p(\mathbf{U}_\ell|\boldsymbol{x}, \boldsymbol{z}) \propto \exp\left(\frac{\varepsilon}{2} \cdot \text{tr}(\mathbf{U}_\ell^\top \mathbf{A}_\ell \mathbf{U}_\ell)\right),$$

which corresponds to a matrix Bingham distribution.

The above presented Gibbs sampler is identical to the one proposed in Section 4.1 in the main text, thus justifying our use of the above-mentioned generative model as an equivalent characterization of the proposed private subspace clustering algorithm. This is perhaps not surprising, as the marginal likelihood model Eq. (27) is exactly the same with the sampling distribution dictated by the exponential mechanism, as shown in Eq. (10) in the main text.

## Appendix E   Private subspace clustering via the SuLQ framework

In this section we introduce a simple iterative subspace clustering algorithm based on the SuLQ framework [2]. Before presenting the algorithm, we first review $k$-plane [3], a straightforward iterative method for subspace clustering:

    1. For each data point $\boldsymbol{x}_i$, compute $z_i = \text{argmin}_{1 \le \ell \le k} d(\boldsymbol{x}_i, \mathcal{S}_k)$.

2. For each cluster $\ell$, let $\mathbf{A}_\ell = \{\boldsymbol{x}_i : z_i = \ell\} \in \mathbb{R}^{d \times n_\ell}$ denote all data points assigned to cluster $\ell$. Update $\mathcal{S}_\ell$ as the linear subspace spanned by the top-$q$ eigenvectors of $\mathbf{A}_\ell \mathbf{A}_\ell^\top$.

3. Repeat step 1 and 2 until convergence.

Suppose the $k$-plane algorithm is run for $T$ iterations. From the pseudocode of $k$-plane, the algorithm needs to query the database $\mathcal{X}$ for $kT$ times, each time asking the following question:

- Given $\mathcal{S}_1, \cdots, \mathcal{S}_k$ and $\ell \in [k]$ as inputs, output the orthonormal basis $\mathbf{U}_\ell \in \mathbb{R}^{d \times q}$ of a $q$-dimensional subspace $\mathcal{S}'_\ell$ such that $\mathcal{S}'_\ell$ best captures $\mathbf{A}_\ell^\top \mathbf{A}_\ell$; i.e., $\|\mathbf{A}_\ell \mathbf{A}_\ell^\top - (\mathcal{P}_{\mathcal{S}'_\ell} \mathbf{A}_\ell)(\mathcal{P}_{\mathcal{S}'_\ell} \mathbf{A}_\ell)^\top\|_2$ is minimized. Here $\mathbf{A}_\ell$ is defined in terms of $(\mathcal{S}_1, \cdots, \mathcal{S}_k)$.

Algorithm 5 is a simple procedure that approximately answers the above question while preserving $(\varepsilon, \delta)$-differential privacy. It is in fact a special case of the SuLQ framework proposed in [2]. The following proposition is immediate.

**Proposition E.1.** *Algorithm 5 is an $(\varepsilon, \delta)$-differentially private algorithm.*

*Proof.* Define $\boldsymbol{b}(\mathcal{X}) = \mathrm{vec}(\mathbf{A}_\ell^\top \mathbf{A}_\ell) \in \mathbb{R}^{d^2}$. Let $\mathcal{X}'$ be an arbitrary database such that $d(\mathcal{X}, \mathcal{X}') = 1$. That is, exactly one column $\boldsymbol{x}$ in $\mathcal{X}$ is replaced by a new column $\boldsymbol{x}'$ in $\mathcal{X}'$. We then have

$$\|\boldsymbol{b}(\mathcal{X}) - \boldsymbol{b}(\mathcal{X}')\|_2^2 \leq \sum_{i,j=1}^d (x_i' x_j' - x_i x_j)^2 \leq 2 \sum_{i,j=1}^d (x_i'^2 x_j'^2 + x_i^2 x_j^2) \leq 4,$$

where the last inequality is due to the constraint $\|\boldsymbol{x}\|_2, \|\boldsymbol{x}'\|_2 \leq 1$. Consequently,

$$\Delta_2 \boldsymbol{b} = \sup_{d(\mathcal{X},\mathcal{X}')=1} \|\boldsymbol{b}(\mathcal{X}) - \boldsymbol{b}(\mathcal{X}')\|_2 \leq 2.$$

The Gaussian mechanism (Theorem A.1, [9]) then suggests that one can release $\boldsymbol{b}$ while preserving $(\varepsilon, \delta)$-differential privacy by calibrating i.i.d. Gaussian noise to $\boldsymbol{b}$:

$$\text{Release} \quad \boldsymbol{b}(\mathcal{X}) + \frac{2\sqrt{2\ln(1.25/\delta)}}{\varepsilon} \cdot \boldsymbol{w},$$

where $\boldsymbol{w}$ is a $d^2$-dimensional standard Normal. The final singular value decomposition step does not affect privacy because differential privacy is close to post-processing. $\qquad\square$

The following proposition is then a direct application of advanced composition [9].

**Proposition E.2.** *Suppose the $k$-plane algorithm is run for $T$ iterations, each iteration querying Algorithm 5 $k$ times with privacy parameters $\varepsilon$ and $\delta$. Then the overall algorithm is $(\varepsilon', \delta')$-differentially private with*

$$\begin{aligned} \varepsilon' &= \sqrt{2kT\ln(1/\delta)}\varepsilon + kT\varepsilon(e^\varepsilon - 1), \\ \delta' &= (kT+1)\delta. \end{aligned}$$

## Appendix F   Concentration theorems

**Theorem F.1** ([44], Theorem 1.2). *Let $\mathbf{A}$ be an $n \times n$ matrices with entries i.i.d. sampled from standard Gaussian distribution. Then there exist absolute constants $c_1 > 0, 0 < c_2 < 1$ such that for every $t > 0$,*
$$\Pr\left[\sigma_n(\mathbf{A}) \leq t\sqrt{n}\right] \leq c_1 t + c_2^n,$$
*where $\sigma_n(\mathbf{A})$ is the least singular value of $\mathbf{A}$.*

**Theorem F.2** (Wedin's theorem; Theorem 4.1, pp. 260 in [45]). *Let $\mathbf{A}, \mathbf{E} \in \mathbb{R}^{m \times n}$ be given matrices with $m \geq n$. Let $\mathbf{A}$ have the following singular value decomposition*

$$\begin{bmatrix} \mathbf{U}_1^\top \\ \mathbf{U}_2^\top \\ \mathbf{U}_3^\top \end{bmatrix} \mathbf{A} \begin{bmatrix} \mathbf{V}_1 & \mathbf{V}_2 \end{bmatrix} = \begin{bmatrix} \mathbf{\Sigma}_1 & \mathbf{0} \\ \mathbf{0} & \mathbf{\Sigma}_2 \\ \mathbf{0} & \mathbf{0} \end{bmatrix},$$

*where* $\mathbf{U}_1, \mathbf{U}_2, \mathbf{U}_3, \mathbf{V}_1, \mathbf{V}_2$ *have orthonormal columns and* $\mathbf{\Sigma}_1$ *and* $\mathbf{\Sigma}_2$ *are diagonal matrices. Let* $\widetilde{\mathbf{A}} = \mathbf{A} + \mathbf{E}$ *be a perturbed version of* $\mathbf{A}$ *and* $(\widetilde{\mathbf{U}}_1, \widetilde{\mathbf{U}}_2, \widetilde{\mathbf{U}}_3, \widetilde{\mathbf{V}}_1, \widetilde{\mathbf{V}}_2, \widetilde{\mathbf{\Sigma}}_1, \widetilde{\mathbf{\Sigma}}_2)$ *be analogous singular value decomposition of* $\widetilde{\mathbf{A}}$. *Let* $\mathbf{\Phi}$ *be the matrix of canonical angles between* range$(\mathbf{U}_1)$ *and* range$(\widetilde{\mathbf{U}}_1)$ *and* $\mathbf{\Theta}$ *be the matrix of canonical angles between* range$(\mathbf{V}_1)$ *and* range$(\widetilde{\mathbf{V}}_1)$. *If there exists* $\delta > 0$ *such that*

$$\min_{i,j} \left| [\mathbf{\Sigma}_1]_{i,i} - [\mathbf{\Sigma}_2]_{j,j} \right| > \delta \ \ and \ \ \min_i \left| [\mathbf{\Sigma}_1]_{i,i} \right| > \delta,$$

*then*

$$\| \sin \mathbf{\Phi} \|_F^2 + \| \sin \mathbf{\Theta} \|_F^2 \le \frac{2 \| \mathbf{E} \|_F^2}{\delta^2}.$$

**Theorem F.3** ([32], Lemma 18, Properties of Gaussian random vectors)**.** *Let* $\boldsymbol{\varepsilon} \sim \mathcal{N}(0, \frac{\sigma^2}{d} \mathbf{I})$ *be a* $d$-*dimensional random Gaussian vector with coordinate-wise variance* $\sigma^2$. *Then the following holds for some fixed* $\boldsymbol{z} \in \mathbb{R}^d$ *and* $t, \rho > 0$:

$$\Pr \left[ \| \boldsymbol{\varepsilon}_i \|_2^2 > (1+t)\sigma^2 \right] \ \le \ e^{\frac{n}{2}(\log(t+1)-t)};$$
$$\Pr \left[ \left| \langle \boldsymbol{\varepsilon}_i, \boldsymbol{z} \rangle \right| > \rho \| \boldsymbol{\varepsilon}_i \|_2 \| \boldsymbol{z} \|_2 \right] \ \le \ 2e^{-\frac{n\rho^2}{2}}.$$

## Footnotes

[5] By definition $R_1 \cup \cdots \cup R_M = S^{d-1}$ and $R_i \cap R_j = \emptyset$ for $i \neq j$.

[6] Note that $|\langle \boldsymbol{y}, \boldsymbol{y}_i^{(\ell)} \rangle| = |\langle -\boldsymbol{y}, \boldsymbol{y}_i^{(\ell)} \rangle|$ by symmetry. So a point far from $\boldsymbol{y}_i^{(\ell)}$ could have large inner product and be connected with $\boldsymbol{y}_i^{(\ell)}$. We take $\tilde{s} = s/2$ to avoid this issue.