[Reviews · NeurIPS 2015]

Submitted by Assigned_Reviewer_1

General comments:

The paper is well written. I would just remove "cryptographic" in the abstract. Not a lot of people are familiar with differential privacy and many of them would get confused with the notion of cryptography in which there exists an algorithm that reverts the "encrypted" data to its original form (I consider that once the data is noisified, its original form is essentially "lost").

Technical comments:

Following the argument in the supplementary information of ref. [30], the authors argue that being able to find a perfect SSC clustering breaches the notion of differential privacy (DP). Agreed. * But * the point is that finding a perfect SSC clustering is just computationally impossible.

Consider the following problem:

Name: SSC approximation. Instance: a set of vectors in R ^ d, integer q

\ geq 1, integer k

\ geq 1, bound b

\ geq 0. Question: does there exist a set of k subspaces of dimension q each such that the subspace clustering objective (as defined by (3) in the paper) is not more than b.

This problem is APX-Complete from reduction from Vertex cover (vertices X, edges E): d is the number of vertices, and for each element of E, create one vector with the indicator variable for the two vertices of X. Fix k = the same as in X3C, b = 1, q = 1. Since SSC solution gives orthonormal bases, solution to SSC approximation gives exactly the indexes of the vertices in X which cover all edges (since b = 1).

Hence, there exists a constant

\ gamma\geq 0 such that if P

\ neq NP, SSC approximation is hard to approximate within 1+ \ gamma.

So any SSC clustering can only guarantee to find an approximation to the optimum. Finally, SSC clustering is no "tougher" for privacy than other models, like the (more) popular barycenter-based clustering algorithms of k-means

/

EM.

However, of course, it is crucial to address SSC clustering in a private framework.

Then, the authors justify the use of a local-sensitivity approach by the fact that global sensitivity would be too high and thus preclude de facto any efficient DP method for the clustering problem. Agreed. * But * then the weaker model somehow imposes to find very efficient algorithms, as otherwise there would be no reason to weaken privacy for (still) inefficient algorithms. From this standpoint, Nissim's approach (ref. [23]) makes clear the fact that the dependence on noise is crucial, and non-trivial to solve even in the SAF framework. In the longer version of their paper, Nissim et al. make this clear (Section 5.1 of www.cse.psu.edu / ~ads22 / pubs / NRS07 / NRS07-full-draft-v1.pdf). Unfortunately, the submission fails to bring to SSC clustering what Nissim et al. could not bring to k-means-type clustering. Actually, the results look comparable from a high-level standpoint. Nissim et al approximate the Wasserstein distance DW (Lemma 5.6 in the longer version) to O (sigma k sqrt(d / n)) while the guarantee in DW for the submission is O (sigma k q log n). Why is it that DW does not decrease at least reasonably when n increases in Theorem 3.6 ? The data being i.i.d., I would expect this to happen.

Furthermore, Lemma 3.5 (the key to Theorem 3.6) is very difficult to appreciate from the qualitative and quantitative standpoints. Drilling down into the conditions, we arrive at the following conclusions: for the probability to be 1-O(1), we *must* have n = O (exp (sqrt(d) / 2)) (first minus parameter in probability), which precludes too large samples. However, looking at the second minus parameter, Jensen's inequality brings n sum exp(-n_l / 400)

\ geq k n exp (-n / (400k)). If k is, say, 10, then to have that last parameter say no more than 1 / 10, we must have exp(n / 4000) \geq 100 n, and thus n \geq 50 000. If d is not large enough, say d \leq 500, the bound is meaningless. And I am not even discussing the three assumptions of the lemma, that are just incredibly hard to capture to figure out a "competency map" of the Lemma that would give sufficient high-level constraints for the "1 - lots_of_things" to be not far from 1 (or even not negative...).

Figuring out such "competency maps" is important and would prevent some "nasty" collateral consequences. For example, Lemma 3.3 is stated with $ \ phi ^ 2 \leq 1 / 802$. However, the constraint cost(hat(C)) \leq a cost (Copt) with a \leq (1-802phi ^ 2) / (800phi ^ 2) imposes, for the right hand side to be larger than 1 (otherwise the statement cannot hold), (1-802phi ^ 2) / (800phi ^ 2) \geq 1, and so... phi ^ 2 \leq 1 / 1602...

The exponential mechanism with Gibbs sampling is original and interesting.

The experiments give interesting results, * But * the following downsides apply: 1) the value of delta is a bit large, considering that delta = 1 / n is the trivial lower bound for which not doing any private mechanism still yields (epsilon, delta) differential privacy. 2) the value of n, d, k are clearly out of the applicable bounds for the theory, so we cannot use the experiments to validate in any way the ideas of the theory. 3) the values of epsilon in the plots is also too large, considering that in most applications, we would have epsilon \leq 1, thus truncating the plots to the [-1, 0] interval on the x-axis. In this case, the approach proposed by the authors clearly gives less compelling results.

Finally, the lack of a conclusion, even short, summarising the pros and the cons, and opening future research directions is not a good thing either.

Summary: the paper is original and its objectives are clear, but it needs some important technical revamping for the theory to be of broad relevance and be technically clear, in particular for the underlying assumptions to be unquestionable. The dependence in noise needs to be improved (or, at least, it must be shown why this large dependence in noise is absolutely necessary) to improve the paper's significance.
Summary: Relevant and non-trivial technical problem, unfortunately treated with technicalities that are hard to appreciate and reveal unsatisfying assumptions and dependences.

Submitted by Assigned_Reviewer_2

The definition of d in eq. (2) are different from that in

Def 2.1 are different. Different symbols should be used.
Summary: This is a light review. The study seems to be technically sound, whereas the method is a straight forward application of sample-aggregate framework to subspace clustering. The conclusion is not that surprising.

Submitted by Assigned_Reviewer_3

The paper presents a differentially private mechanism for subspace clustering. Three mechanisms are proposed: one is based on the idea of local sensitivity from earlier work, one employs stochastic data and one is based on an exponential mechanism. For the first and second mechanisms, the main effort is to establish a utility result via what is essentially a stability argument that rests on somewhat-standard assumptions about the input. The third approach makes use of a Gibbs sampling technique to provide a more practical (but without guarantees) heuristic for the differentially private clustering release.

Quality: The work here is theoretically solid, and it's highly nontrivial to prove the stability results, because the underlying metric is a subspace metric which is tricky to work with. But this is the main set of results: the privacy guarantees come essentially "for free" from earlier work by Nissim et al: that guarantee requires a stability property that the authors establish.

The exponential mechanism is also interesting as well as the Gibbs sampler provided for it. But the experiments aren't really conclusive one way or anothe, especially in regard to the SuLQ method.

Clarity: the paper is overall quite well written and easy to understand. I might quibble a little at how impenetrable the theorem statements are, but that's a matter of taste.

Originality: There's a lot of work that has gone into the stability proofs. I didn't get a clear sense even from the supplementary material as to whether the proofs were tedious but straightforward, or had new ideas. I think the authors could have helped out a little here.

Significance: As a personal (biased) note, I'm not sure that subspace clustering is the best platform for differentially private mechanisms.

Summary: A good paper that analyzes differentially private mechanisms for clustering. But it's really a stability result in disguise. '

Submitted by Assigned_Reviewer_4

The authors first analyse two algorithms with provable privacy and utility guarantees for subspace clustering (in the agnostic and stochastic settings). They then propose a private subspace clustering method that uses an exponential mechanism, and then provide an alternating Gibbs sampling scheme, that in the second step requires sampling from a matrix-Bingham distribution, for which they provide an efficient sampling scheme.

They then provide numerical results on synthetic and real datasets, showing that their algorithm simultaneously maintains privacy whilst performing well in higher dimensional settings. The authors have gone to great lengths to explain their methods (primarily in the supplementary material) and the contribution here appears to be highly significant. On the negative side, the paper is lacking any concluding comments to tie the theoretical and empirical results together, and there are many small grammatical errors (some detailed below) and a few notational issues. Finally, there 5 separate references to unpublished (arxiv) papers - if there are published versions of these work they should be replaced if possible.

Detailed comments:

A table of notation in the supplementary material would be useful! L60: along our analysis -> during our analysis L61: and exact clustering -> and an exact clustering L71: both work to -> both works to L99: for a generic function f, more often than not f^2(x) means f(f(x)) and does not mean (f(x))^2, however I presume that this is what d^2(x) means here - could you clarify? L124: In definition 1, is d(X,Y) = 1 required or would d(X,Y) \leq 1 suffice? L147: ruins utility -> ruins the utility L149: introduces -> introduced L156: what is \omega? L158: what is \Lambda? L180: In addition, utility -> In addition, the utility L181: In following -> In the following L194: well-seperation -> well-seperated L195: as above L199: \Delta_{k-1}, \Delta_{k, -} and \Delta_{k, +} -> \Delta^2_{k-1}, \Delta^2_{k, -} and \Delta^2_{k, +} L255: implies well-separated -> implies a well-separated L256: noise level -> the noise level L259: what is s? L286: guarantee -> guarantees L296: samples -> sample L304: release clustering -> release a clustering L316: update -> updates L364: matrix -> matrices L371: Gibbs -> for Gibbs L417: Such preprocessing -> Such a preprocessing L418: ambient -> the ambient L419: projection -> projections L502: U and U' should be in bold? L539: (supplementary) In the proof of Lemma B.4., where is the connection to Proposition A.3?

Summary: An interesting paper with theoretical and algorithmic developments, this seems like a strong contribution.

Author Feedback
Author rebuttal: We thank the reviewers for their helpful comments.

To R#10:

- All f^2(x) mean (f(x))^2 in this paper. We apologize for this notation confusion.

- In Def. 1: here d(.,.) is defined as the number of columns in which two databases differ and is always a non-negative integer. When d(X,Y)=0 we have X=Y and Eq.(4) trivially holds.

To R#2: we thank the reviewer for a thoughtful and detailed review, which raises a few excellent points. Our response is as follows:

- comparison between subspace clustering and general center-based clustering: we agree that perfect agnostic subspace clustering is computationally intractable. However, with some additional data assumptions it is possible to get computationally efficient perfect clustering algorithms [1,2,3]. This gives subspace clustering an additional dimension compared to general center-based clustering problems like k-means.

- DW does not decrease in n: this is indeed a very good point. We believe this is an artifact of our analysis: at the final step (after correct clustering is obtained) we apply SVD perturbation bound over the original data matrix for each subspace, which is too loose under the stochastic model. A better idea is to consider the MLE estimator of the Probabilistic PCA model [4], which is similar to the stochastic setting in our paper. The MLE estimator involves estimation of noise magnitude and soft-thresholding of singular values, which is known to be computationally efficient and statistically consistent. This argument will give us the desired dependency over n (proportional to 1/sqrt(n)) in the final DW bound.

- Numerical issues with theoretical bounds: we thank the reviewer for paying attention to this point. The general interpretation is, when the ambient dimension d is not too small (at least Omega(log^2 n)), the theorem holds with high probability. Our analysis was geared towards real-world big datasets like face images and network hop counts data that have large number of data points and quite high ambient dimension (over a thousand).

- We realize that our experiments cannot corroborate theoretical findings due to numerical constants. However, the experiments are designed to compare proposed private subspace clustering algorithms in a practical setting, on both synthetic and real-world datasets. We do have a few interesting observations from the experiments. For example, SAF performs quite poorly (which agrees with previous empirical findings on SAF k-means) and SuLQ degrades when dimension d is not small, since it releases the covariance matrix at each iteration. We also remark that the exponential mechanism is an (epsilon,0)-private mechanism and hence the setting of delta does not affect it.

- We will add a short conclusion summarizing pros/cons of the proposed algorithms and connections between theoretical and empirical findings.

To R#4:

- Our experiments show that the proposed Gibbs sampling algorithm is comparable to the SuLQ framework under low-dimensional (small d) settings. However, when d is not smal (center and right figures), the performance of SuLQ is much weaker. This is because SuLQ releases a d x d matrix per iteration and the quality of the released matrix worsens as d increases.

- Our stability results of subspace clustering are inspired by previous analysis of the Lloyd's algorithm for k-means clustering. However the extension to subspace clustering is not trivial and we incorporate ideas from the subspace clustering literature as well, such as the TSC algorithm.

- Though differential privacy has been little investigated for subspace clustering before, we do think this is an important problem. As explained in the second paragraph of the introduction, subspace clustering is the method of choice for many applications, specially involving images, that contain sensitive information and the clustering results need to be released with care.

To R#6: our paper may be a little bit technical for a light review. We will continue improving the paper to make it more accessible to a broader audience.

To R#7: we prove non-trivial stability results for subspace clustering (which are required by sample-and-aggregate framework) under two different settings. This is perhaps the main theoretical contribution of this paper. On the practical side, we provide a gibbs sampling algorithm based on the exponential mechanism and demonstrate its effectiveness on both synthetic and real-world datasets. We believe our results make a significant contribution to the field of subspace clustering.

References:
[1]. Sparse Subspace Clustering: Algorithm, Theory and Applications. TPAMI, 2013.
[2]. A Geometric Analysis for Sparse Subspace Clustering with Outliers. Annals of Statistics, 2011.
[3]. Robust Subspace Clustering via Thresholding. Arxiv:1307.4891.
[4]. Mixtures of probabilistic principle component analyzers. Neural Computation, 1999.